# Biochemical Modification of Poly-Vinyl-Alcohol-Based Bioplastics with *Citrus* By-Product to Increase Its Food Packaging Application

**DOI:** 10.3390/ijms26199470

**Published:** 2025-09-27

**Authors:** Giuseppe Tancredi Patanè, Stefano Putaggio, Davide Barreca, Annamaria Russo, Annamaria Visco, Cristina Scolaro, Rosalia Maria Cigala, Francesco Crea, Salvatore Abate, Federica De Luca, Silvana Ficarra, Ester Tellone, Giuseppina Laganà, Antonella Calderaro

**Affiliations:** 1Department of Chemical, Biological, Pharmaceutical and Environmental Science, University of Messina, 98166 Messina, Italy; giuseppe.patane@studenti.unime.it (G.T.P.); stefano.putaggio@studenti.unime.it (S.P.); rosaliamaria.cigala@unime.it (R.M.C.); francesco.crea@unime.it (F.C.); salvatore.abate@unime.it (S.A.); federica.deluca@unime.it (F.D.L.); sficarra@unime.it (S.F.); ester.tellone@unime.it (E.T.); giuseppina.lagana@unime.it (G.L.); anto.calderaro@gmail.com (A.C.); 2Engineering Department, University of Messina, 98166 Messina, Italy; annamaria.visco@unime.it (A.V.); cscolaro@unime.it (C.S.); 3Institute for Polymers, Composites and Biomaterials, CNR-IPCB, Via Paolo Gaifami 18, 95126 Catania, Italy

**Keywords:** biodegradable bioplastic, circular economy, active packaging, antioxidant activity, functionalized polymeric structures, food packaging, polyphenols

## Abstract

The necessity to produce new biodegradable polymeric materials, to overcome the economic model, based on the linear economy, and to apply the circular economy model is a global problem. As a result, components unutilized derived from industrial processes are becoming increasingly valuable and useful to create new materials. This work focuses on the production of bioplastics based on poly (vinyl) alcohol (PVA) that have been modified with flavonoid fraction, liquid fraction obtained after digestion with cellulase and pectinase, and the solid material remaining after enzyme treatment, obtained from *Citrus bergamia* by-product (the so-called “pastazzo”). This last one is an almost completely unutilized product, although it is a potential rich source of biological active compounds. Enzymatic and non-enzymatic green extraction protocol have been employed to separate the different fractions and to make it more suitable to functionalize the PVA, suppling new properties to the bioplastics in a dose-dependent manner. Morpho-functional analysis was conducted by SEM, XRD, colorimetry, UV–visible and ATR-FTIR spectroscopy. Regarding optical properties, the obtained results show that transparency of the film in terms of light transmittance (T%) for PVA alone is very high, but when functionalized it had a reduced T%. From the data obtained, the functionalized films acquire antioxidant activity, as well as good mechanical properties, making them good candidates for biodegradable packaging for preserving the shelf life of different fruits and vegetables as confirmed by the food fresh-keeping test performed on apple samples.

## 1. Introduction

Nowadays, it is very difficult to think of anything in our routine life that we use directly/indirectly that is not made of plastic. The fact that this material is now ubiquitous in our lives is evidenced by the increase in plastic production by about 230 times since 1950 [1]. European countries alone produced 29 million tons of plastic only in 2018. The problem with this high production is that about 80 percent of it, in the classic linear economy model, ends up as waste that is discarded, generating high disposal costs and, most importantly, pollution for the environment [2]. The circular economy model stems from the need to change the linear life of commercial products, generally used and thrown away, into a circular life, in which used and consumed objects are modified, improved and reintegrated into the market, so as to reduce disposal costs and generate new economic value [3]. In 2014, the European Commission decided to adopt for the first time the circular economy model (CE), in which the 4Rs approach “reduce, reuse, recycle, and recover” emphasizes that in this model, every market-generated by-product must be reused and reintroduced into the market. This model becomes even more important when one considers that about 90% of all waste generated by the plastics market comes from non-biodegradable and fossil-based plastics, which over time generate environmental pollutants for us, land and sea animals [4]. In this context, researchers are trying to find new biodegradable materials that offer different solutions to the industrial market [5,6]. Among the different materials, polyvinyl alcohol (PVA) is a synthetic but biodegradable polymer, obtained by polyvinyl acetate hydrolysis, which has gained the attention of the industrial market and researchers due to its good thermal and mechanical properties [7,8]. In the literature, there are several scientific papers in which PVA is used as a starting material, in which other components are added to this polymer in order to reduce its final concentration, consequently its cost, and to improve its elasticity and water resistance, such as glycerol and carbohydrates or starch; however, its ability to interact with samples containing water is still an open challenge [8,9,10]. In addition, it is important to consider that PVA also behaves as an excellent matrix capable of incorporating biologically active compounds that can impart new properties to bioplastics, such as antioxidant and antimicrobial, in order to functionalize food packaging [11]. Overall, all these needs, from a circular economic perspective, can be satisfied by utilizing all the useful molecules offered by the natural world, whose products are often used as foods, fruit juices or for the production of essential oils, producing tons of waste that can still offer economic value from a nutraceutical and industrial point of view. For example, in Italy, and particularly in Reggio Calabria, there is an important production of bergamot (*Citrus bergamia*), of which three varieties are known (Castagnaro, Fantastico and Femminello), whose fruits are mainly used for the production of essential oils and *Citrus bergamia* juice. In recent years, chemical characterization of these fruits has highlighted how they are a rich source of polyphenols and flavonoids, which together give them important biological activities, such as antioxidants, anti-cancer, anti-inflammatory, antimicrobial, and neuroprotective [12,13,14,15]. After the fruit is deprived of its juice and essential oils, which are useful for cosmetic and nutraceutical actions, about 60 percent of the fruit’s weight remains, which is considered waste; it is mainly represented by *Citrus* peels, contains a lot of pectins and flavonoids and is also called “pastazzo” [16]. Although recently, some *Citrus* peel has been employed in the production of a “packaging system” for protecting perishable fruits because they are rich, in cellulose, pectin and polyphenols; at the same time, it still lacks the utilization of the main by-product (“pastazzo”) derived from agro-industrial utilization [17,18,19].The aim of our work was to make all the components of the “pastazzo” suitable for full utilization of these waste products, in accordance with the European Agency’s circular economy objective, through a green extraction technique, according to green chemistry, to reintroduce flavonoids and the polysaccharide matrix, still present in the ‘pulp’ of *Citrus bergamia*, into the economic system, valorizing them by functionalizing new PVA-based bioplastics in a different way, so as to modify their optical and mechanical properties and give them new antioxidant properties useful for food packaging.

## 2. Results and Discussions

*Citrus bergamia* “pastazzo” is a by-product obtained from the *Citrus* industry that is still particularly rich in bioactive and interesting compounds that can be employed to build a circular economic model and avoid the vicious cycle of its poor utilization and consequent cost for its disposal with potential impact on the environment. Its utilization can be considered as pivotal for an effective bioeconomy strategy for the rural development and, in fact, there are increasing investments in *Citrus* waste in the main producer country [20,21,22]. Within this scope, “pastazzo” management is one of the major issues for citrus processors due to the high costs incurred, for instance, for pre-treatments before its disposal [23,24]. Every year, ~98 million tons of *Citrus* fruits (mainly oranges, lemons, limes, grapefruits and tangerines) are processed by the food industry, and ~50–60% of them become waste, the so-called “pastazzo” composed of water (75–85%), mono- and disaccharides (6–8%), and a limited level of oils in the peel waste [25]. The essential oils, where present, also represent a potential risk to the environment if they are not appropriately disposed of [26,27]. Several attempts have been made to valorize this by-product such as extraction of pectin, dietary fibers and essential oils (particularly D-limonene) and fermentation processes to produce biogas and ruminant feeding [28,29]. In line with European Union (EU) directives and models proposed by the United Nations for appropriate management of by-products derived from the agro-industrial sector and their reduction by 2030, the by-products selected for this study were Sicilian *citrus* product [30].

### 2.1. Extraction and Characterization of Polyphenols Matrix from Citrus bergamia “Pastazzo”

The extraction of polyphenols present in *Citrus bergamia* “pastazzo” were initially performed using the simulation program AGREE (Analitical GREenEss calculator, (Michigan Engineering, 1221 Beal Ave. Ann Arbor, MI 48109-2102) [31]. It allowed the selection of solvent and extraction methods with the release of the best index. Subsequently, extraction and identification were carried out. The starting point was to analyze and to compare the classical reference systems for extraction (Soxhlet extraction) with the methods present in the literature, which give the highest yields (50% dimethylformamide in water), and to program greener systems. In Figure 1 is depicted the green index obtained with the simulation program. The results of the simulation clearly indicated the possibility to utilize green solvents and ultrasound to achieve more eco-friendly processes than traditional ones. 

The conditions and techniques we used for the simulation with the AGREE program were then utilized in laboratory to extract the polyphenols and separate and quantify them by RP-HPLC-DAD. In Figure 2 is depicted the chromatogram of the polyphenols identified at 280 nm and 325 nm obtained from *Citrus bergamia* “pastazzo”. The comparison between the two wavelengths allowed us to carry out an initial screening and to distinguish between flavonoid components belonging to the subclass of flavones and flavanones, both of which are characterized by two absorption bands (referred to as Band I and II); however, in flavanones the band at 280 nm is much more pronounced than at 325 nm. In particular, the HPLC-DAD separation, recorded at 278 nm, let us identify the following compounds: vicenin-2 (1); lucenin-2 4′-methyl ether (2); eriocitrin (3); isovitexin (4); neoeriocitrin (5); rhoifolin (6); naringin (7); chrysoeriol 7-O-neohesperidoside (8); neohesperidin (9); neodiosmin (10); and bergapten (11).

The identification has been supported also by comparison with literature data and co-elution with commercially available standards. The analysis of the quantitative data clearly shows that one of the adopted green methods allows us to obtain higher yields than traditional techniques and other methods present in the literature. In particular, the extraction by sonication in ethanol:H_2_O (70:30, *v*:*v*), at the temperature of 70 °C for 30 min had the highest yield (~1.00 ± 0.37% of the starting dried materials). In general, this extraction technique in the range of a temperature of 40–70 °C is between ~1.25 and 1.36 times higher than extraction by maceration in DMF:H_2_O (50:50, *v*:*v*) at RT for 30 min, which has the highest yield among the other methods used.

### 2.2. Bioplastics Characterizations

#### 2.2.1. ATR-FTIR Analysis

Morpho-functional analysis of the bioplastics was performed by means of ATR-FTIR spectroscopy. In the first spectrum, the typical absorption bands of PVA are characterized by the presence of the bands at 3249 cm^−1^ (O–H stretching), 2936 cm^−1^ (asymmetric stretching of CH_2_), 2906 cm^−1^ (symmetric stretching of CH_2_), 1642 cm^−1^ (due to water absorption), 1416 cm^−1^ (CH_2_ bending), 1325 cm^−1^ (δ (OH), rocking with CH wagging), 1141 cm^−1^ (shoulder stretching of C–O) (crystalline sequence of PVA), 1083 cm^−1^ (stretching of C=O and bending of OH) (amorphous sequence of PVA), 915 cm^−1^ (CH_2_ rocking) and 822 cm^−1^ (C–C stretching). As can be seen from the graph, the modification of the starting materials with the flavonoid fraction and the digested fraction obtained following treatment with pectinase and cellulase does not induce the appearance of a new band due to the formation of covalent linkage with PVA, indicating a dispersion of utilized materials to functionalize the new bioplastic, but mainly the appearance of a broad and intense band at about 1600–1650 cm^−1^. This can be attributed to the greater vibration energy of the carbonyl group present in the identified flavonoids, utilized to functionalize the materials and the water molecules bound to the digested fraction obtained from *Citrus* waste with cellulase and pectinase. The treatment with pectinase and cellulase is fundamental because pectin is the jelly-like matrix which helps cement plant cells together with other cellular biomolecules, such as cellulose fibrils, making the materials more accessible to the creation of the new materials. The further modification of the material with the addition of the remaining solid fraction after the separation of the flavonoids and the enzymatic digestion brings a significant change in the ATR-FTIR spectrum of the obtained materials. In fact, as can be seen in Figure 3, the presence of cellulose characteristic bands can be observed in the region of 800–1630 cm^−1^. Furthermore, broad bands in the region between 1600 and 1700 cm^−1^ can be observed, due to the bending mode of water molecules absorbed to the films and whose intensity is proportional to the increasing amount of O–H groups. A characteristic fingerprint region can be found between 1000 and 1200 cm^−1^, displayed by defined bands attributable to the C–O–C asymmetric stretching of the β-glycosidic bond at 1135 cm^−1^, C–C bonds of the monomer units of polysaccharide at 1070 cm^−1^, C–O–C pyranose ring stretching vibration around 1047 cm^−1^, and C–O symmetric stretching of the primary alcohol at 1020 cm^−1^. There are also some changes in the bands at 1138 cm^−1^ and 1083 cm^−1^ influencing the sequence with crystalline and amorphous structures that are directly related to the modification of the PVA, in accordance with the obtained data that are presented in the following sections.

#### 2.2.2. SEM Analysis

The SEM images highlight different surface morphologies, ranging from relatively smooth surfaces (e.g., Figure 4, sample 1) to highly rough surfaces characterized by the formation of ridges or well-defined structures, often distributed non-uniformly and exhibiting a tendency to form agglomerates (e.g., Figure 4, sample 4). The presence of flavonoid fraction and the digested material obtained following treatment with pectinase and cellulase appears to significantly influence the surface morphology, with a general increase in roughness. Furthermore, as the addition of the remaining solid fraction after the separation of the flavonoids and the enzymatic digestion concentration increases, a decrease in sample porosity is observed (Figure 4, samples 3 and 4). Additionally, this last addition seems to promote the formation of distinct phases, such as the observed ridges, or more complex structures that ultimately influence the overall surface morphology.

The SEM images, further processed into 3D surface plots (Figure 5), highlight the effect of citrus-derived fractions on the topography of PVA-based films. The pure PVA film (sample 1) exhibited a relatively compact and homogeneous surface with moderate roughness and limited irregularities, consistent with a calculated surface porosity of 2.93%. Upon incorporation of the flavonoid-rich fraction and the liquid digested fraction (sample 2) and 0.5 mg solid fraction (samples 3), the overall topography appeared more irregular, with an increased number of surface protrusions and valleys. Interestingly, quantitative image analysis (with ImageJ 1.54g) revealed that surface porosity decreased to 0.014% and 0.008% for samples 2 and 3, respectively, suggesting that these fractions compact or fill open pores despite increasing roughness. The lowest value of 0.005% was observed for the film containing the enzymatically digested solid residue (sample 4), despite its more pronounced surface roughness and heterogeneity.

This apparent discrepancy can be explained by the fact that the incorporation of citrus-derived solids tends to partially fill or compact open surface pores, thereby reducing the 2D porosity detected by threshold-based image analysis, while simultaneously generating localized agglomerates that emerge as surface peaks and enhance roughness. As a consequence, samples 2–4 show reduced planar porosity but increased morphological irregularities. These features correlate with the functional properties described in the preparation of the bioplastics: films with flavonoid and liquid digested fractions (samples 2) display improved tensile strength and barrier properties, whereas the excessive roughness and local discontinuities introduced by the maximum addition of solid residue (sample 4) act as structural modifier, leading to diminished mechanical resistance and higher opacity. The combined 3D surface and porosity analysis provide quantitative confirmation of the link between morphological modifications and the balance between strengthening and weakening effects in functionalized PVA films. Roughness parameters (Ra, Rq) were extracted from SEM line profiles using ImageJ and expressed in arbitrary intensity units, serving only as comparative indicators. The trend observed was sample 4 < sample 1 < sample 2 < sample 3. Interestingly, SEM line profile analysis indicated lower Ra/Rq values for sample 4, despite its irregular surface morphology. This apparent discrepancy arises from the method: single or limited line profiles capture average intensity variations along selected traces and may overlook sparse protrusions evident in 3D reconstructions. In fact, sample 4, with very low porosity (0.005%), displays a morphology of few large agglomerates surrounded by smooth areas; as a result, line-based Ra appears lower, whereas area-based and 3D analyses reveal pronounced local irregularities.

#### 2.2.3. XRD Analysis

The XRD patterns of the as-prepared samples are presented in Figure 6. The pure PVA diffractogram exhibits several broad and poorly defined peaks, indicative of a poorly ordered crystalline structure, typical of semicrystalline polymers. Notably, the prominent crystalline reflection at 2θ = 20° is characteristic of PVA, corresponding to reflections from the (101) plane1. As shown in Figure 6, the characteristic PVA peak persists in all samples, suggesting that the addition of other reactants has a minimal impact on the underlying structure. For instance, while the addition of flavonoid fraction and the digested material obtained following treatment with pectinase and cellulase appears to alter the intensity of existing diffraction peaks, indicating an interaction between flavonoids and PVA that influences the polymer’s molecular organization, no new peaks were detected. Conversely, increasing the remaining solid fraction (obtained after the separation of the flavonoids and the enzymatic digestion) leads to changes in the diffractograms (new peaks at 2θ = 13° and 24.4°), suggesting interactions between PVA and all the added components, resulting in subtle alterations to the overall material structure.

The morphological and structural modifications observed by SEM and XRD correlate directly with the functional properties of the films. Pure PVA exhibited the characteristic reflection at 2θ ≈ 20° with a crystallinity index (CI) of 65%. After incorporation of flavonoid and liquid digested fractions (sample 2) and the lowest amount of solid fraction (sample 3), the CI decreased to 41% and 45%, while simultaneously reducing surface porosity to 0.014% and 0.008%. This structural rearrangement promoted a more amorphous matrix and limited diffusion pathways, thereby enhancing chain mobility and enabling stronger intermolecular interactions between PVA and citrus-derived polysaccharides. As a result, these samples exhibited significantly improved tensile strength (up to 79.56 ± 2.21 MPa) and elongation (up to 384.16%), together with an increased contact angle (from ~15° in neat PVA to ~22°), reflecting improved barrier properties against aqueous environments. Conversely, the film containing the solid residue (sample 4) partially recovered crystallinity (CI = 60%) due to the presence of cellulose II X1, as demonstrated by the new peaks at 2θ ≈ 13° and 24° with index (110) and (020), respectively. The coexistence of crystalline cellulose domains and a heterogeneous PVA network explains the increased surface irregularities observed in SEM, with surface porosity reduced to 0.005% but roughness markedly increased. These morphological discontinuities acted as stress concentrators, leading to decreased mechanical resistance (20.14 ± 2.24 MPa; elongation 45.77%) and reduced transparency due to enhanced light scattering [32].

### 2.3. Thickness and Mechanical Properties of Functionalized Bioplastics

The data reported in Table 1 and the stress–strain graph of Figure 7 show that the addition of both flavonoid fraction and the digested material obtained following treatment with pectinase and cellulase and remaining solid fraction (obtained after the separation of the flavonoids and the enzymatic digestion) increases the film thickness, as expected. The thickness doubles from about 32 μm to a value between 67 and 79 μm. The result of the mechanical tests indicates that the addition of flavonoid fraction and the digested material obtained following treatment with pectinase and cellulase solutions to the PVA sample increases the mechanical resistance and the deformability of the material (from about 35 MPa to about 79 MPa, and from about 119% to about 384%, respectively). On the other hand, if the remaining solid fraction (obtained after the separation of the flavonoids and the enzymatic digestion) is also added to the PVA, the mechanical resistance and the deformability of the material drastically worsen (to a value of about 20 MPa and of 45%, respectively). In terms of stiffness, the addition of both components decreases Young’s modulus, from about 1221 MPa to a range of about 526–623 MPa. It is possible that the addition of remaining solid fraction (obtained after the separation of the flavonoids and the enzymatic digestion) to functionalized bioplastics acts as a plasticizer, which increases the distance and mobility between the functional groups during casting, decreasing the strength of the material, as also reported in other works [33]. Therefore, the analysis of the mechanical performance of PVA is improved by the addition of the flavonoid fraction and the digested material obtained following treatment with pectinase and cellulase, but it is decreased by the further addition of remaining solid fraction (obtained after the separation of the flavonoids and the enzymatic digestion). The addition of the flavanol fraction and the liquid fraction obtained after digestion with cellulase and pectinase decreases the stiffness of the material (from 1221 MPa to 526 MPa) because it lubricates the macromolecular chains, facilitating their sliding and therefore their deformability. However, the addition of 0.5 g and 1.0 g of the solid material remaining after the enzymatic treatment increases stiffness again to 623 MPa and 594 MPa due to the stiffening effect caused by the presence of the solid fraction, which hinders the sliding of the macromolecular chains within the chemical structure of the bioplastic. These results must be evaluated according to the ultimate purpose for which the bioplastics are designed, so as to have the right balance between strength and flexibility and to obtain the complete utilization of the waste product. The modification of the starting material is also evident in the analysis of the contact angle values (about 15° degrees, for PVA film alone to almost 22° degrees after its modification with the utilizing materials). This clearly indicates a decrease in the hydrophilicity of the starting material, making its utilization suitable with samples containing water.

Polybutylene succinate (PBS) is an example of a bioderived and biodegradable plastic of scientific interest, as it is used commercially for various applications [6,34]. Considering its tensile mechanical parameters (modulus: 328 MPa, breaking strength: 37 MPa and elongation at break: 354%), we can conclude that pure PVA (sample A) is stiffer and less deformable than PBS, since its elastic modulus is approximately 1200 MPa and its breaking strength is approximately 118% (Table 1). However, the mechanical behavior of PBS is more similar to sample B, which extends its deformability by approximately 384%. This indicates that the plasticizing effect of the flavonoids and the liquid fraction obtained after digestion with cellulose/pectinase makes PVA more similar to PBS. Therefore, it would be interesting to conduct a future study on the PBS biopolymer matrix modified with *citrus* by-products.

### 2.4. Optical Properties and Appearance

The PVA control appears transparent, while following functionalization all bioplastics acquire a yellow–orange color. The samples of bioplastics functionalized with flavonoid fraction and the digested material obtained following treatment with pectinase and cellulase appear homogenous in terms of color, suggesting that all the polyphenols and liquid fraction are well distributed in the matrix and are compatible with PVA. The bioplastics functionalized with the addition of both flavonoid fraction and the digested material obtained following treatment with pectinase and cellulase and remaining solid fraction (obtained after the separation of the flavonoids and the enzymatic digestion) show a much greater grade of roughness if compared with the other samples. In detail, the colorimetric analyses conducted clearly show that L* decreases in all functionalized samples, while a* and b* increase, especially the latter increases from 2.77 for PVA alone to 47.46 following the addition of the liquid fraction. It can also be seen from Table 2 that bioplastics functionalized with the solid fraction appear slightly darker than those functionalized with the liquid fractions alone.

All bioplastics were analyzed in terms of their transparency and opacity, parameters considered essential for plastics used in food packaging industries, as they indicate the amount of UV–visible light that can pass through bioplastics and possibly induce changes in food matrices. As shown in Table 3, PVA alone is a very transparent material with a T% value of 89.75%, indicating that almost all the incident light can pass through the material, while after functionalization, T% levels drop to a minimum value of 38.29%. These results are also confirmed by the literature, where there is strong evidence that the addition of components with UV-absorbing aromatic groups results in a lower T% than PVA alone, which is a translucent material [9,35,36]. Contrary to this, when we consider opacity, it goes from a value of 1.56 for PVA alone to 6.88 for the most functionalized bioplastics with all the components obtained from the waste product.

### 2.5. Antioxidant Power of Functionalized Bioplastics

In order to confirm that the bioplastics we have produced fit into the concept of “active packaging”, as established by European regulations (Regulation (EC) No. 450/2009 (29 May 2009), and that they are therefore capable of preserving the life of foodstuffs, depending on their antioxidant activity, we have performed the most common abiotic assays to test their real potential, such as DPPH, FRAP, Cuprac and ABTS. In fact, it has been widely demonstrated that the production of free oxygen radicals (ROS), especially in foods with a high unsaturated fatty acids content, can alter the organoleptic properties of food, leading to its premature loss and non-consumption [37]. As can be seen from the results (Figure 8), in all the assays, PVA alone shows no antioxidant activity, which is also confirmed by the literature [38]. Following functionalization, the biofilms exhibit marked antioxidant capacity. This new activity was first demonstrated against the stable DPPH radical, in which the addition of flavonoid fraction and the digested material obtained following treatment with pectinase and cellulase obtained from *Citrus bergamia* “pastazzo” leads to a maximum reduction of about 57% of the radical present in the solution. Continuing with the ET/HAT-based assays, the antioxidant power was also confirmed against the green ABTS radical, where there is a marked discoloration, hence scavenger activity, when the radical comes in contact with the functionalized biofilms; there is a residual of about 17% radical following interaction with bioplastic functionalized with flavonoid fraction and the digested material obtained following treatment with pectinase and cellulase, while this value increases to about 35% following functionalization with 0.1% flavonoid fraction, liquid fraction obtained after digestion with cellulase and pectinase and 0.5 or 1.0 g of the solid material remaining after enzyme treatment.

Even in the FRAP colorimetric assay, in which the development of blue coloration is dependent on the reduction of Fe^3+^ to the ferrous ion Fe^2+^, it can be seen that PVA alone does not exhibit any antioxidant activity, whereas the functionalized biofilms express a strong reducing power, which in the case of the functionalized biofilm reaches about 88% of scavenging activity. Finally, the antioxidant power was also confirmed by the Cuprac assay. Again, as can be easily seen from Figure 8d, it can be observed that PVA alone has no activity, whereas in the presence of functionalization, the results are comparable to 5.69 ± 0.23 μM trolox equivalents. The results reported so far confirm how the functionalization of PVA with waste obtained from *Citrus bergamia* ‘pastazzo’ gives new antioxidant activity, which can protect the food they contain and increase its preservation, shelf life, and preserve its organoleptic properties. The polyphenols previously identified through RP-HPLC-DAD analysis and incorporated inside the matrix make these biofilms fully part of the concept of “active packaging”, in which the incorporated compounds act according to their direct release to the food or the environment surrounding it. Furthermore, these results also fully satisfy the need to fit into the circular economy model, as everything that was considered simple waste fully fits into a production process, in which the characteristics of a simple material like PVA are implemented. In addition, these results also confirm that it is not necessary to use synthetic antioxidants such as propyl gallate, 2-butylatedhydroxytoluene and 3-tert-butyl-4-methoxyphenol, as was performed in the past in the food industry, but it is cheaper and safer to use natural antioxidants obtained from the natural world and, in particular, from food waste products, as proposed from other and our research groups [39,40].

### 2.6. Migration Test of Polyphenols in Food Simulants

Using UV–visible spectroscopy (Beckman DU 640 spectrophotometer; Harbor Boulevard, Fullerton, CA, USA) and indirect determination methods, the levels of polyphenols released within the different food simulants were quantified. As can be seen from the graphs in Figure 9, the amount of polyphenols released at different time intervals (0, 2, 4, 6, 8 and 24 h) in H_2_O, 10% ethanol, 50% ethanol and 3% acetic acid was analyzed for each created bioplastic. The results clearly indicate that the greatest release for each sample in which polyphenols are present occurs in contact with the lyophilic 50% (*v*/*v*) ethanol solution and a minor amount in the 10% (*v*/*v*) EtOH solution, confirming what is reported in the literature [41,42,43].

These results strongly indicate that our bioplastics have a greater affinity for fatty foods, with which there is a faster release of polyphenols, while for less fatty or acidic foods there is a more gradual and slower release, respectively. Also, from an extractive point of view, it has been shown that *Citrus bergamia* polyphenols are more easily extracted in the presence of hydro-alcoholic or 100% alcoholic solutions [44,45,46]. These results, together with those obtained for the antioxidant tests, confirm that our bioplastics made from *Citrus* by-product have not only new antioxidant activity but also the ability to release them in the surrounding environment and potentially in the food they encounter, improving their shelf life and organoleptic properties. Furthermore, if these compounds are released into the surrounding environment or foodstuffs, this means that they are able to act as food fortifiers and increase the quality of the packaging elements with bioactive compounds, able to promote human wellness and, on the other side, to increase the intrinsic biodegradability of the material once it is released in the environment [47,48]. On the other side, the same speech cannot be taken into account for acidic matrices or polar environments.

### 2.7. Food Fresh-Keeping Test

The results obtained so far, which show that the bioplastics produced fall into the category of “active food packaging”, have also been confirmed by experiments conducted on real food matrices, in particular on *Malus domestica*. In fact, to see if these biofilms are indeed suitable for our intended purpose, we produce rectangular food packaging bags by the heat-sealing method, as can be seen in Figure 10A. After packing the apple pieces inside the bioplastics, we analyzed the oxidation state and the antioxidant power of the samples at 0, 24 and 48 h of incubation by colorimetric analysis and antioxidant assay.

In particular, the colorimetric tests presented in Table 4 show that the apple not protected by any bioplastic, after 24 and 48 h, registers a notable decrease in L* (71.38 to 38.20) and a notable increase in a* (−1.61 to 15.12) and b* (19.24 to 34.82), indicating a decrease in the clarity of the sample and the color shift towards orange/yellow, typical of the oxidation and browning phenomenon in the apple. In the case of the apple protected with PVA alone, the colorimetric results are better 24 h after the start of the process, while they are very similar with respect to the control after 48 h. The results evidenced that the non-functionalized bioplastic fails to protect the sample already after one day from the start of the process, while in the case of the apple protected with the functionalized bioplastic the changes are almost completely negligible after 24 h, with little changes after 48 h (Table 4), showing that the functionalized bioplastics can improve the shelf life of the food matrix in the period considered. These colorimetric analyses, in which the color change as a function of oxidation phenomena is observed, are also confirmed by spectrophotometric analyses, in which we tested the antioxidant potential of the samples based on DPPH assay (Figure 10B). As can be seen in the graph, the apple samples used for the analysis also have good antioxidant power, capable of scavenging the presence of the stable DPPH radical by about 72.5%. After 24 and 48 h from the start of the process, there is a remarkable decrease in the apple’s antioxidant power corresponding to 56.3 (loss of antioxidant potential of about 22.5%) and 17.0% (loss of antioxidant potential of about 77.0%), respectively. A similar behavior can also be seen for the apple samples packed with only PVA (Figure 10B). Different behavior is recorded for all apple samples packed within the functionalized bioplastics, as the scavenger activity of the apple samples is almost completely superimposable with the one of apples without incubation (at 0 time) after 24 h. After 48 h of incubation in the functionalized bioplastics there is a reduction in antioxidant potentials, corresponding to around 50.0%, which is always clearly superior to the one with the unprotected apples or with PVA alone (Figure 10B).

## 3. Materials and Methods

### 3.1. Reagents and Standard Solutions

Polyvinyl alcohol (PVA, Mw 89,000–98,000, 99% hydrolysis), methanol, ethanol, acetic acid, HPLC-grade acetonitrile, sodium acetate, sodium phosphate dibasic, potassium phosphate monobasic, ammonium acetate, sodium carbonate, ethylenediaminetetraacetic acid (EDTA), copper (II) chloride, 2,2′-azinobis (3-ethylbenzothiazoline-sulfonic acid) diammonium salt (ABTS), 2,2-diphenyl-1-picrylhydrazyl (DPPH), 2,4,6-tris(2-pyridyl)-ttriazine (TPTZ), 6-Hydroxy-2,5,7,8-tetramethylchroman-2-carboxylic Acid (Trolox), Folin–Ciocalteu phenolic solution were purchased from Merck (Darmstadt, Germany). Ammonium peroxydisulfate was purchased from BioRad (Hercules, CA, USA). Dimethylformamide (DMF) was supplied by Carlo Erba (Milano, Italy). All other chemicals and solvents used in this study were of analytical grade unless specified.

### 3.2. Collection of Pastazzo and Extraction Optimization of Polyphenols

The fresh *Citrus bergamia* pastazzo was kindly donated in August by a local company near Reggio Calabria. The fresh pastazzo was homogenized with a waring blender and passed through a sieve with 100 μm mesh, aliquoted in portions of 100.0 g frozen, lyophilized (main drying: −23 °C 0.770 mbar for 20 h; final drying: −76 °C −0.001 mbar for 2 h) and stored at −20 °C until utilization.

### 3.3. Polyphenol Extraction and Characterization by Reverse Phase High-Performance Liquid Chromatography Coupled with Diode Array (RP-HPLC-DAD)

Fixed amount of “pastazzo” (prepared as described above) was extracted with different solvents. The simulation program AGREE has been used to investigate and improve previous classical methods used for the quantification and characterization of polyphenol fraction. The simulated and later-performed extraction conditions were as follows: soxhlet in H_2_O; soxhlet in ethanol; maceration in H_2_O at RT for 30 min; maceration in ethanol–H_2_O (70:30, *v*:*v*) at RT for 30 min; maceration in DMF:H_2_O (50:50, *v*:*v*) at RT for 30 min; sonication in H_2_O at 40, 50, 60 and 70 °C, respectively, for 30 min and sonication in ethanol–H_2_O (70:30, *v*:*v*) at 40, 50, 60 and 70 °C, respectively, for 30 min. To perform the real analysis, 10.0 mg of “pastazzo” were extracted in the condition described above with a ratio of 1:100 (*w*:*v*), filtered with filter paper and concentrated until reaching the final volume of 10.0 mL. The identification of present polyphenols was performed by utilizing a Shimazu Reverse Phase–Diode Array Detection–High Performance Liquid Chromatography (Shimadzu Ltd., Canby, OR, USA) with injection loop of 20.0 L. The column was a BioDiscovery C18 of 250.0 mm × 4.6 mm i.d., 5.0 μm and equipped with a 20.0 mm × 4.0 mm guard column. The temperature was set at 30 °C and flowrate at 1.0 mL/min. The separation was performed utilizing a linear gradient of acetonitrile in H_2_O as mobile phase. The gradient was: 0–15 min (5–20% of acetonitrile), 15–20 min (20–30% of acetonitrile), 20–35 min (30–100% of acetonitrile), 35–40 min (100% of acetonitrile), 40–45 min (100–5% of acetonitrile) and 45–55 min (5% of acetonitrile). The chromatograms were recorded at 278 and 325 nm and UV–visible spectra of each peak were registered between 200 and 450 nm. The identification of the compounds was performed according to retention time, UV spectra, and co-elution with authentication standards. Quantitative analysis was carried out by integration of the areas of the peaks from the chromatogram at 278 nm for flavanones and 325 nm for flavones and comparison with calibration curves obtained with the known concentration of a commercially available standard (0.1–10.0 mg/L).

### 3.4. Enzymatic Digestion of Pastazzo

After removing the polyphenols and analyzing their composition, the pulp residue was subjected to enzymatic extraction to facilitate the release and solubilization of pectins and polysaccharides. In detail, 10.0 g of the residue was dissolved for 24 h in 400 mL of acetate buffer pH 5.0 in order to hydrate it and subjected to enzymatic digestion with 0.132 g of pectinase and 2.080 g of cellulase and left under agitation for four days at a temperature of 37 °C. At the end of the incubation time, the sample was centrifuged at 4000 rpm for 10 min and the supernatant was filtered through filter paper. The two phases were then separated, and the supernatant (liquid fraction obtained after digestion with cellulase and pectinase) and precipitate (solid material remain after enzyme treatment) were stored at −20 °C until further use.

### 3.5. PVA Based Bioplastic Production

For the preparation of the bioplastics, we used the solvent casting method, as reported in the literature, with minor modifications (incubation at 25 °C with 5.0% humidity for 72 h) in order to functionalize the bioplastics [49]. Briefly, we solubilized 2.6 g of PVA in 50 mL of liquid fraction obtained after digestion with cellulase and pectinase and left it for one hour at 90 °C under stirring until a homogeneous solution was obtained. Subsequently, while keeping the temperature at 45 °C and the solution in agitation, the polyphenols were added in such a way as to obtain a final concentration of 0.1% for each bioplastic, in the absence or in the presence of 0.5 or 1.0 g solid material remaining after enzyme treatment, in order to confer varying degrees of strength to the bioplastics. Thus, we prepare three types of materials: PVA plus 0.1% flavonoid fraction and liquid fraction obtained after digestion with cellulase and pectinase; PVA plus 0.1% flavonoid fraction, liquid fraction obtained after digestion with cellulase and pectinase and 0.5 g of the solid material remaining after enzyme treatment; and PVA plus 0.1% flavonoid fraction, liquid fraction obtained after digestion with cellulase and pectinase and 1.0 g of the solid material remaining after enzyme treatment. As a control we prepared PVA alone, as described above, without the addition of the functionalizing fraction and solubilizing the PVA powder in water.

### 3.6. Film Characterization

#### 3.6.1. Attenuated Total Reflection-Infrared Spectroscopy

Attenuated Total Reflection-Infrared Spectroscopy (ATR-FTIR) was used to assess whether the functionalization of PVA alone with flavonoids and various carbohydrate concentrations resulted in the modification or creation of new functional groups. All spectra were acquired with the Thermo Scientific spectrophotometer model iS50 ATR (Nicolet™ iS50 FTIR Spectrometer. Waltham, MA, USA, Stati Uniti) and were performed in the 4000–600 cm^−1^ region, with a resolution of 4 cm^−1^, for a total of 32 scans per sample.

#### 3.6.2. Scanning Electron Microscopy

The morphology of functionalized bioplastics was investigated by a scanning electron microscope (Phenom ProX Desktop instrument. Waltham, MA, USA, Stati Uniti) with a 10 kV acceleration voltage in BSE mode. The images were captured between 200 and 50 µm and by varying the magnification in the range of 330–1650×.

#### 3.6.3. X-Ray Diffractometer

Crystallographic phases of the tested bioplastics were identified using a Bruker D8-Advance X-ray diffractometer (XRD) (Milano, Italy) with Cu-Kα radiation (λ = 1.54186 Å). XRD analysis in the 2θ range of 10° to 72° was employed to determine the crystallographic phases and structures of the synthesized materials.

#### 3.6.4. Optical Properties

To measure the different optical properties of PVA-based bioplastics before and after functionalization with carbohydrates and flavonoids, we calculated for each sample the transmittance, using the method reported by do Nascimento and co-workers [33]. In detail, we cut out several pieces of equal size and placed them in a spectrophotometer solid object holder and acquired the transmittance in percent (%T) of each sample at 600 nm. The transmittance value was changed into absorbance (Abs) and then used to obtain opacity values based on the thickness (mm) of each sample using the following formula:$$Opacity=\frac{Abs}{Thickness}$$

Functionalization, especially with flavonoids, also involves colorimetric changes, which were recorded using the Optis colorimeter. We set a visual angle of 10° and an illuminant value of D65. As a reference, we used a white plate. In our samples, L* varied from 0 (black) to 100 (white), a* varied from green (−) to red (+) and b* varied from blue (−) to yellow (+); we calculated the overall color differences after functionalization using the following equation:$$∆E=\sqrt{{\left(∆L\right)}^{2}+{\left(∆a\right)}^{2}+{\left(∆b\right)}^{2}}$$

### 3.7. Mechanical Properties

Static tensile tests were performed on the dog-bone-shaped specimens according to the standard procedure of ASTM D 638-03. The universal dynamometer used was a Lloyd LR10K with a 0.5 kN load cell (Elis–Electronic Instruments & Systems S.r.l., Rome, Italy). The preload value was 1.00 N. The speed of the moving crosshead was 1 mm/min. The tests were performed at 25 °C and 30% relative humidity (RH). The mechanical parameters considered for this test were the following: the tensile strength at failure (Lr [N]), the stress and strain at failure (σr [MPa] and εr [%], respectively), and the stiffness or Young’s modulus (E [MPa]). The values reported in Table 1 represent an average value calculated on eight specimens for each type of material. The contact angle (θw) was evaluated by the sessile drop method using a DMs-401 Contact Angle Meter (KYOWA) Agaram Industries, 73 Nelson Road, Aminjikarai, Chennai 600 029, India. The instrument measures the contact angle of a 2 μL drop of deionized water on the horizontal surface of the biofilm. The contact angle value is derived from the following equation [50]:θω=2cot(2hd)

Biofilm thickness was assessed with a digital thickness gauge (SAMA Tools SA8850) (SAMA Italia, Viareggio, Italy). Calculation method: on each circular biofilm sample (diameter 8 cm) a map of 6 × 6 = 36 points is graphically identified in which the vertical probe is positioned to measure the thickness. Once this is achieved, the average values of all measurements are calculated. Statistical analysis was obtained with the software Prism 8.0.2. It allows us to obtain data as mean ± SD (±Standard Deviation) at a significance level of *p* < 0.05. For the normality and lognormality of the data, the D’Agostino and Pearson test was used, while the Brown–Forsythe test was used for the homogeneity of variance test.

### 3.8. Antioxidant Capacity of Bioplastics

In order to understand how the functionalization of these new bioplastics can add new properties, the most-used antioxidant in vitro assays were performed. In detail, since assays to test antioxidant activity are classified mechanistically into hydrogen atom transfer (HAT)-based and electron transfer (ET)-based assays, we performed the 2, 2,2-diphenyl-1-picrylhdrazyl (DPPH), 2,20-azino-bis(3-ethylbenzothiazoline-6-sulfonic acid) (ABTS) as both ET-based and HAT-based assays, and the Cuprac and ferric reducing power (FRAP) as exclusively ET-based [51]. For all antioxidant activity experiments, the bioplastics were cut in squares with an equal weight of approximately 0.0048 g.

#### 3.8.1. Inhibition of the Free Radical 2,2-Diphenylpyrylhydrazyl Radical (DPPH) Assay

To perform the free and stable radical 2,2-diphenyl-1-picrylhydrazyl (DPPH) assay, we followed the protocol reported by Barreca and co-workers [52]. In detail, a solution of DPPH was prepared to have a final concentration of 80.0 μM. In the cuvette, small dilutions were made with methanol to have a final absorbance value below 1.0. Subsequently, pieces of bioplastic were placed in the cuvette with 1.0 mL of the working solution (DPPH/methanol) and shaken for 30 s. After 30 min of incubation, readings were taken at 517 nm, using the Varian Cary 50 UV-Vis spectrophotometer. The antioxidant activity, expressed as I (%), was calculated using the following formula:$$I\left(\%\right)=\frac{\left[\left(Ac-As\right)\right]}{Ac}\times 100$$
where Ac represents the absorbance of the control and As the absorbance of the sample. All tests were carried out in triplicate and the results were expressed as mean ± standard deviation (SD).

#### 3.8.2. Ferric-Reducing Antioxidant Power (FRAP) Assay

To perform the assay, we followed the protocol reported by Barreca et al. with minor modifications [53]. FeCl_3_ 20.0 mM, 2,4,6-Tris(2-pyridyl)-S-triazine (TPTZ) 10.0 mM in HCl 40.0 mM and acetate buffer 300.0 mM, pH 3.6 were prepared. The working solution was prepared by mixing acetate buffer, TPTZ and FeCl_3_ in a ratio of 10:1:1 and stored at room temperature until use. The samples were cut so that they all had the same weight and incubated in a cuvette with 750 μL of working solution for 4 min. Absorbance was recorded at 595 nm using a spectrophotometer (BECKMAN DU 640, Brea, CA, USA). All experiments were carried out in triplicate and the results were expressed as % inhibition of radical.

#### 3.8.3. Cupric Reducing Antioxidant Capacity (Cuprac)

The Cuprac assay was performed as reported in the literature [54]. In detail, CuCl_2_ 10.0 mM, Neocuproine (Nc) 7.5 mM and ammonium acetate solutions (NH_4_Ac) 1.0 M, pH 7.0 were prepared. Next, the working solution was prepared by mixing CuCl_2_ aqueous solution, ammonium acetate solution and neucoproine in equal ratios until a blue color was obtained. The absorbance of this solution was monitored at 450 nm and kept below 1.0. Subsequently, the bioplastic samples were incubated in 1.0 mL of this solution, and the colorimetric changes were recorded after 30 min of incubation at room temperature. All experiments were performed in triplicate, and the antioxidant activity of the bioplastics was expressed as trolox activity.

#### 3.8.4. ABTS

The antioxidant activity of the functionalized bioplastics against the stable and free radical 2,2′-azino-bis(3-ethylbenzothiazoline-6 sulphonic) acid (ABTS) was analyzed following the method reported in the literature [55]. Briefly, for the preparation of the cationic radical, a solution of 7.0 mM ABTS and 10.0 mM ammonium peroxydisulfate were mixed together in phosphate buffer solution pH 7.2 overnight. From the stock solution, appropriate dilutions were made so that the absorbance value at 734 nm was less than 1.0. Subsequently, the cut bioplastic samples were placed in cuvettes with 1.0 mL of diluted ABTS solution and spectrophotometer readings were taken after 6 min. Each analysis was performed in triplicate and the results were expressed as % cation radical elimination.

### 3.9. Release and Quantification of Polyphenols in Different Food Simulants

As required by the European standard EN. (2002), we investigated the behavior of our new bioplastics when in contact with solutions that mimic the nature of different foods. In this case, we worked as reported in the literature, with minor modifications, in order to quantify the release of polyphenols in the different food simulants [37,56]. In detail, we cut and prepared bioplastics into equal pieces of approximately (1.0 × 1.0 cm^2^) and different food simulants solutions, 100% water (*v*/*v*), 10% ethanolic solution (*v*/*v*), 50% (*v*/*v*) ethanolic solution and acidified solution with 3% acetic acid (*v*/*v*), respectively, in order to mimic the migration of different components between the bioplastics and various foods and to see the migration of the different components. After immersing the bioplastics in these solutions, samples were taken at regular intervals to see the release and perform the quantification of total polyphenols by Folin–Ciocalteu method [57]. All experiments were performed in triplicate.

### 3.10. Packaging and Food Fresh-Keeping Test

In order to demonstrate whether the bioplastics that we produced and functionalized can be considered useful for food packaging purposes, we modified them into food bags, chose apples (*Malus domestica*, variety STARK) as the food matrix and carried out various analyses as reported in the literature to analyze their shelf life at different time intervals [49,58]. All the operations reported were carried out under a fume hood in order to prevent contamination from interfering with the data obtained. The apple was cut into equal-sized pieces 1.0 × 1.0 × 0.5 (L × W × H) cm with a professional stainless-steel cutter with 1/2- and 1/4-inch blades, obtaining a weight of 0.95 ± 0.12 g for each apple sample. Subsequently, the samples were stored with or without the produced bioplastics. To demonstrate how bioplastics are able to protect apple samples from oxidation and deterioration over time, we analyzed the color change in the samples using an Eoptis CLM-194 colorimeter (Trento, Italy), selecting the CIE L*a*b* color space and analyzing the color change as a result of oxidative browning. After these analyses, the samples were diluted 1:1 with PBS, homogenized with Potter homogenizer and centrifuged at 10,000 rpm. The supernatant obtained was subsequently used to measure the antioxidant power of the samples against the stable DPPH radical at 517 nm, as reported before [52].

### 3.11. Statistical Analysis

Data are expressed as means ± standard deviation (S.D.). Statistical analyses were performed by one-way analysis of variance (ANOVA). The significance of the difference from the respective controls for each experimental test condition was assayed by using Dunnett’s test for each paired experiment. A *p* < 0.05 was considered statistically significant. For mechanical data, Prism 8.0.2 statistical software was used for the statistical analysis. Data are reported as mean ± SD (±standard deviation) at a significance level of *p* < 0.05. The Shapiro–Wilk test was used for normality and lognormality tests of data, and the Brown–Forsythe test for homogeneity of the variance test. Since all data used in this study satisfied these two tests, two-way analysis of variance (ANOVA) with Tukey’s post hoc test was performed to evaluate the statistical significance of the differences between the groups (significance level: 0.05).

## 4. Conclusions

The management of plastic materials and the production of new biodegradable ones is an open field with a lot of challenges to solve, with the potential to have great development for both the design and production of new polymeric materials. In this paper, we explored this potential and reported the manufacture of biodegradable class III bioplastic (synthetic material produced from petrochemicals but biodegradable) utilizing as a starting material PVA and functionalizing it with a *Citrus* industrial by-product, which falls in the field of circular economy. The functionalized produced materials show interesting results acquiring new properties (such as antioxidant activity) and decreasing hydrophilicity, without drastically changing mechanical features; as far as the optical properties are concerned, they create materials that are more able to protect the packed products from radiation. The direct application of the produced materials on fresh fruit samples show us that it is able to preserve apple samples’ freshness and to avoid oxidation, supporting their potential in utilizing them as bio-based packaging systems for the extension of food shelf life, representing also a circular system model able to utilize (after easy, not expansive, and green procedures) all the components present in the by-product to produce a high value new material, meeting the goal of the 2030 European agenda. Moreover, the bio-packaging system produced in this work can open new roads to reutilize all the components derived by *Citrus* agro-industrial by-products and holds promise for reducing plastic pollution and improving food shelf life.

## Figures and Tables

**Figure 1 ijms-26-09470-f001:**
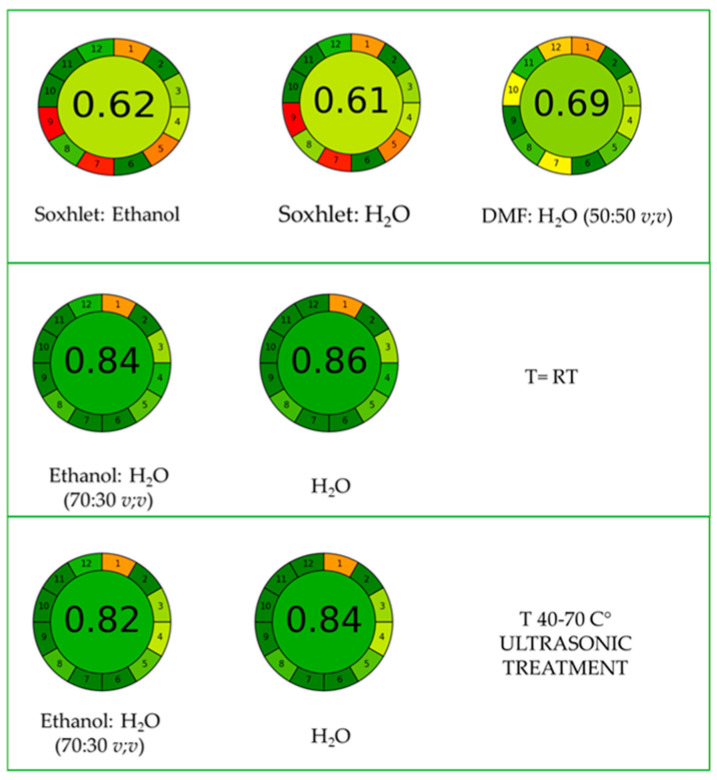
Comparison of the different extraction methods to be used in the experiment, carried out using the AGREE prediction program. The colors, red-yellow-green, indicate performance at each stage of the process. The less is chemically “green” the process, the redder it appears.

**Figure 2 ijms-26-09470-f002:**
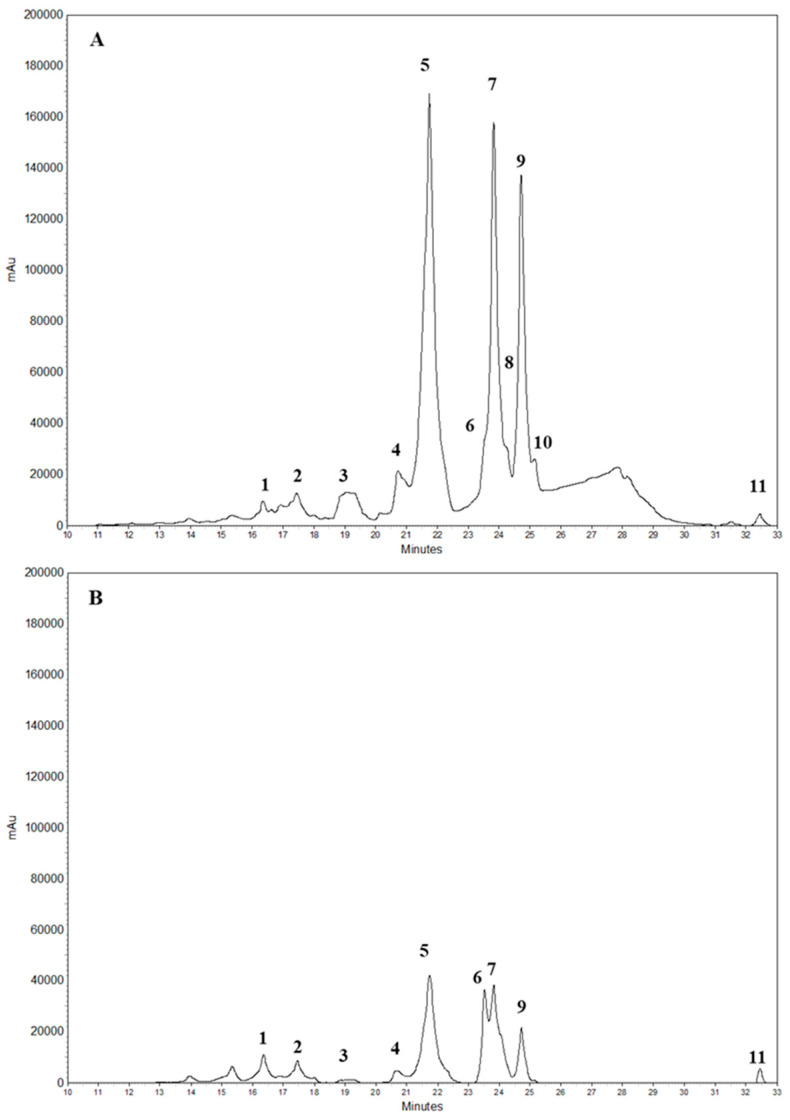
Representative RP-HPLC-DAD chromatogram separation of the flavonoids obtained from *Citrus bergamia* pastazzo recorded at 278 and 325 nm, obtained by ethanolic extraction (70:30, *v*:*v*) of *Citrus bergamia* by-product after industrial extraction of the juice and essential oils. Components 1–11 were identified as follows: vicenin-2 (1); lucenin-2 4′-methyl ether (2); eriocitrin (3); isovitexin (4); neoeriocitrin (5); rhoifolin (6); naringin (7); chrysoeriol 7-O-neohesperidoside (8); neohesperidin (9); neodiosmin (10); bergapten (11).

**Figure 3 ijms-26-09470-f003:**
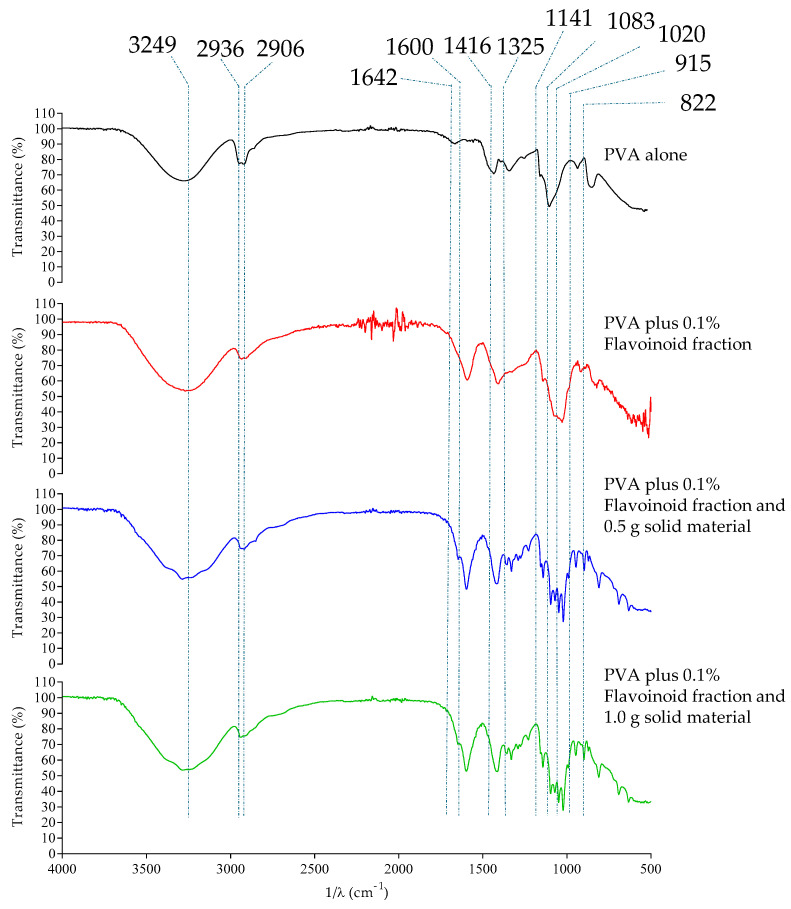
Representative ATR-FTIR spectra of new PVA-based bioplastics produced by the addition of increasing *Citrus bergamia* by-product obtained during our experimentation. (

) PVA alone; (
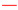
) PVA plus 0.1% flavonoid fraction and liquid fraction obtained after digestion with cellulase and pectinase; (
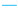
) PVA plus 0.1% flavonoid fraction, liquid fraction obtained after digestion with cellulase and pectinase and 0.5 g of the solid material remaining after enzyme treatment; (
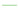
) PVA plus 0.1% flavonoid fraction, liquid fraction obtained after digestion with cellulase and pectinase and 1.0 g of the solid material remaining after enzyme treatment. Band assignment: 3249 cm^−1^ (O–H stretching); 2936 cm^−1^ (asymmetric stretching of CH_2_); 2906 cm^−1^ (symmetric stretching of CH_2_); 1643 cm^−1^ (water absorption); 1416 cm^−1^ (CH_2_ bending); 1325 cm^−1^ (δ (OH); rocking with CH wagging); 1141 cm^−1^ (shoulder stretching of C–O) (crystalline sequence of PVA); 1083 cm^−1^ (stretching of C=O and bending of OH) (amorphous sequence of PVA); 1020 cm^−1^ (C–O symmetric stretching of the primary alcohol); 915 cm^−1^ (CH_2_ rocking); 822 cm^−1^ (C–C stretching).

**Figure 4 ijms-26-09470-f004:**
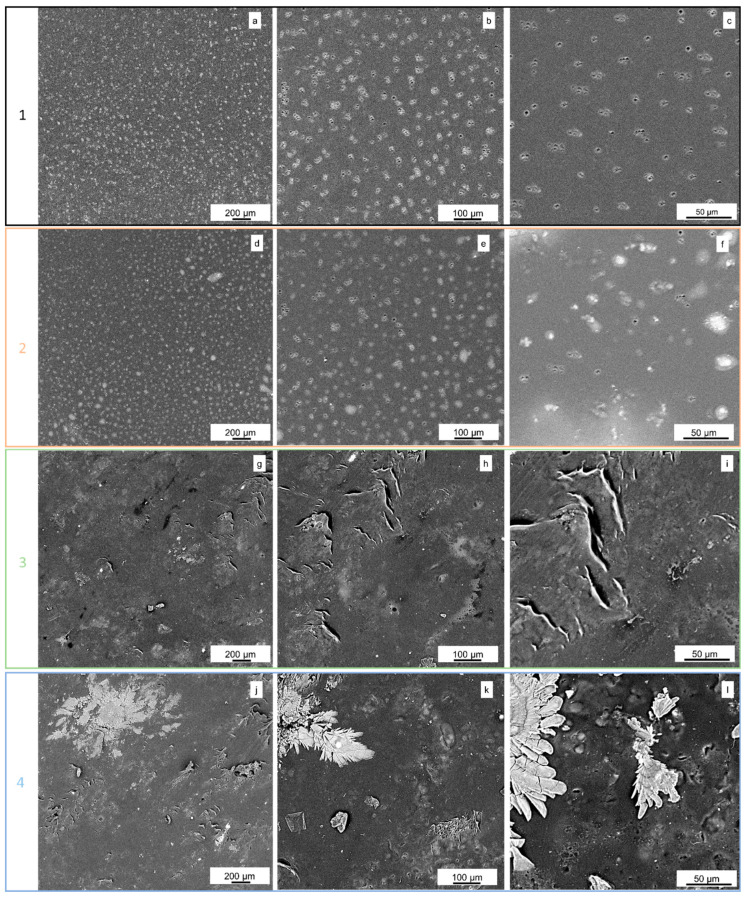
SEM images of new PVA-based bioplastics produced by the addition of increasing *Citrus bergamia* by-product obtained during our experimentation. 1—(**a**–**c**) PVA alone; 2—(**d**–**f**) PVA plus 0.1% flavonoid fraction and liquid fraction obtained after digestion with cellulase and pectinase; 3—(**g**–**i**) PVA plus 0.1% flavonoid fraction, liquid fraction obtained after digestion with cellulase and pectinase and 0.5 g of the solid material remaining after enzyme treatment; 4—(**j**–**l**) PVA plus 0.1% flavonoid fraction, liquid fraction obtained after digestion with cellulase and pectinase and 1.0 g of the solid material remaining after enzyme treatment, at 200, 100 and 50 µm, respectively.

**Figure 5 ijms-26-09470-f005:**
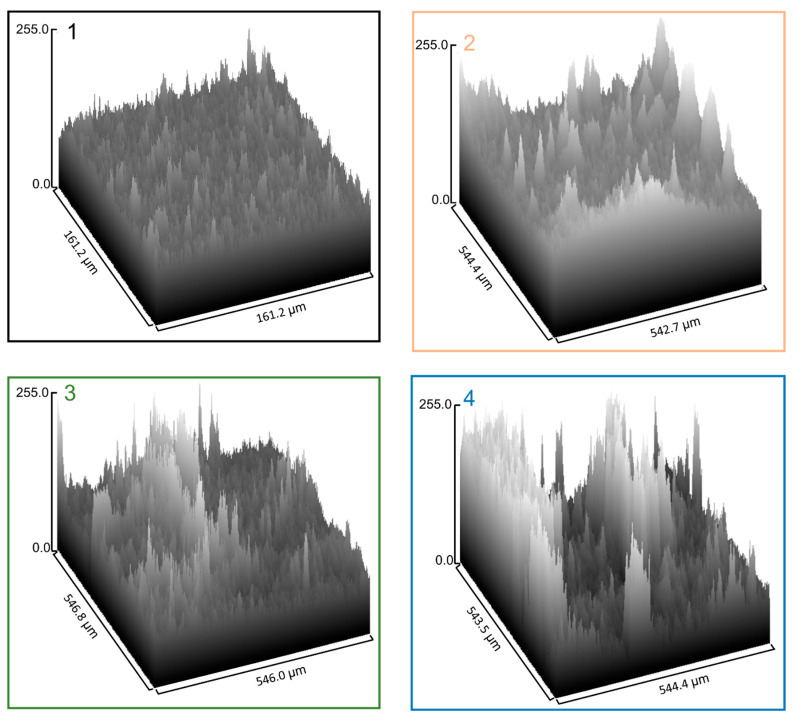
3D surface plots of PVA-based films obtained from SEM micrographs after ImageJ processing: (**1**) PVA alone (sample 1); (**2**) PVA plus 0.1% flavonoid fraction and liquid fraction obtained after digestion with cellulase and pectinase (sample 2); (**3**) PVA plus 0.1% flavonoid fraction, liquid fraction obtained after digestion with cellulase and pectinase and 0.5 g of the solid material remaining after enzyme treatment (sample 3); (**4**) PVA plus 0.1% flavonoid fraction, liquid fraction obtained after digestion with cellulase and pectinase and 1.0 g of the solid material remaining after enzyme treatment (sample 4). A progressive decrease in surface porosity (2.93%, 0.014%, 0.008% and 0.005%, respectively) is observed, together with an increase in surface roughness and heterogeneity, particularly in sample 4, where the incorporation of solid residue leads to pronounced topographical irregularities.

**Figure 6 ijms-26-09470-f006:**
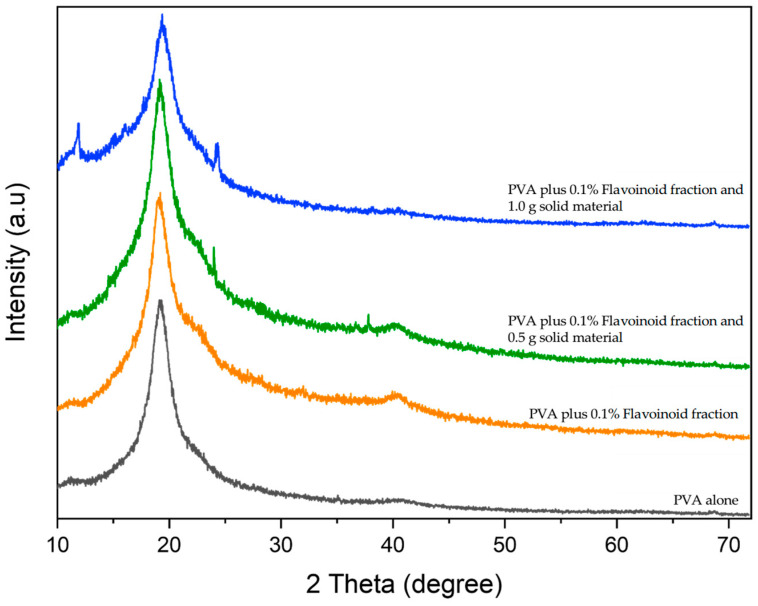
The XRD spectra of the prepared bioplastics. PVA alone (black line); PVA plus 0.1% flavonoid fraction and liquid fraction obtained after digestion with cellulase and pectinase (orange line); PVA plus 0.1% flavonoid fraction, liquid fraction obtained after digestion with cellulase and pectinase and 0.5 g of the solid material remaining after enzyme treatment (green line); (4) PVA plus 0.1% flavonoid fraction, liquid fraction obtained after digestion with cellulase and pectinase and 1.0 g of the solid material remaining after enzyme treatment (blue line).

**Figure 7 ijms-26-09470-f007:**
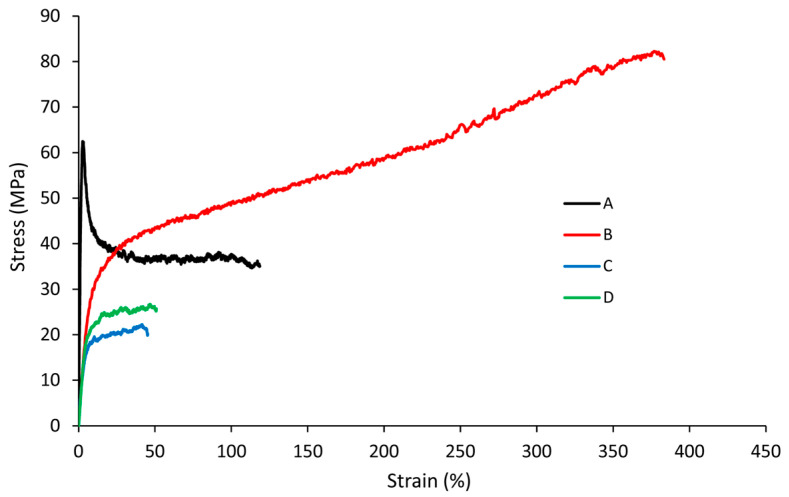
Representative stress–strain curves of new PVA-based bioplastics produced by the addition of increasing *Citrus bergamia.* PVA alone (black line); PVA plus 0.1% flavonoid fraction and liquid fraction obtained after digestion with cellulase and pectinase (red line); PVA plus 0.1% flavonoid fraction, liquid fraction obtained after digestion with cellulase and pectinase and 0.5 g of the solid material remaining after enzyme treatment (blue line); PVA 0.1% flavonoid fraction, liquid fraction obtained after digestion with cellulase and pectinase and 1.0 g of the solid material remaining after enzyme treatment (green line).

**Figure 8 ijms-26-09470-f008:**
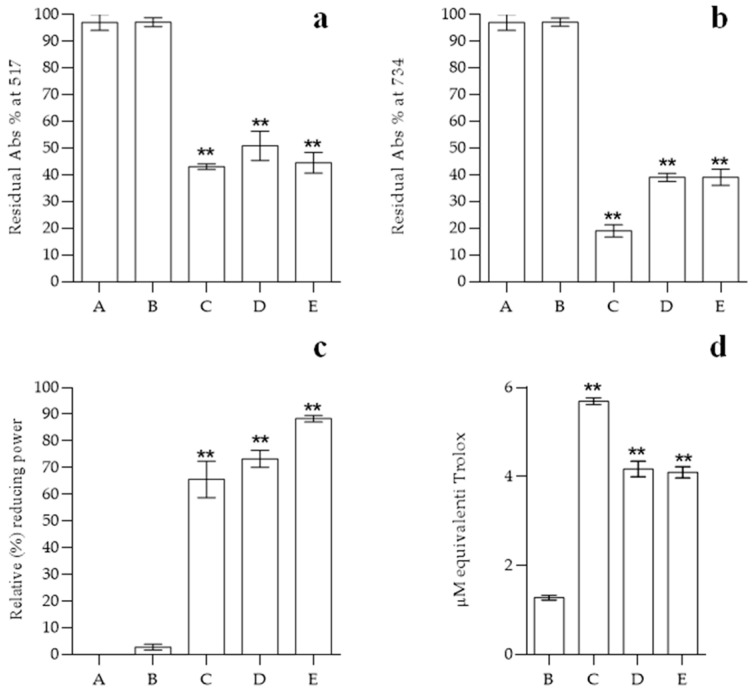
Evaluation of antioxidant activity of PVA-based bioplastics produced by the addition of increasing *Citrus bergamia* by-product obtained during our experimentation. (**a**) DPPH assay; (**b**) ABTS assay; (**c**) ferric reducing power (FRAP) assay; (**d**) CUPRAC. The letters in the different graph indicate the following: A, control sample; B, PVA alone; C, PVA plus 0.1% flavonoid fraction and liquid fraction obtained after digestion with cellulase and pectinase; D, PVA plus 0.1% flavonoid fraction, liquid fraction obtained after digestion with cellulase and pectinase and 0.5 g of the solid material remaining after enzyme treatment; E, PVA plus 0.1% flavonoid fraction, liquid fraction obtained after digestion with cellulase and pectinase and 1.0 g of the solid material remaining after enzyme treatment. ** *p* < 0.001.

**Figure 9 ijms-26-09470-f009:**
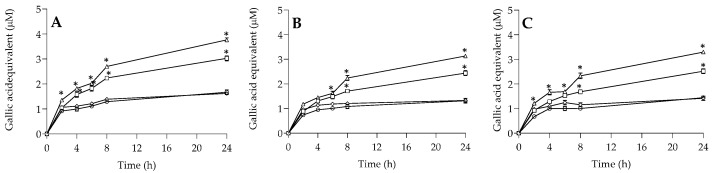
Release for short- and long-term migration of polyphenols from PVA films in different food simulants. (**A**) PVA plus 0.1% flavonoid fraction and liquid fraction obtained after digestion with cellulase and pectinase; (**B**) PVA plus 0.1% flavonoid fraction, liquid fraction obtained after digestion with cellulase and pectinase and 0.5 g of the solid material remaining after enzyme treatment; (**C**) PVA plus 0.1% flavonoid fraction, liquid fraction obtained after digestion with cellulase and pectinase and 1.0 g of the solid material remaining after enzyme treatment. (○) H_2_O; (□) ethanol 10%; (△) ethanol 50%; (◊) acetic acid 3%. Asterisk (*) indicates significant difference with respect to the samples in H_2_O at a *p* < 0.05.

**Figure 10 ijms-26-09470-f010:**
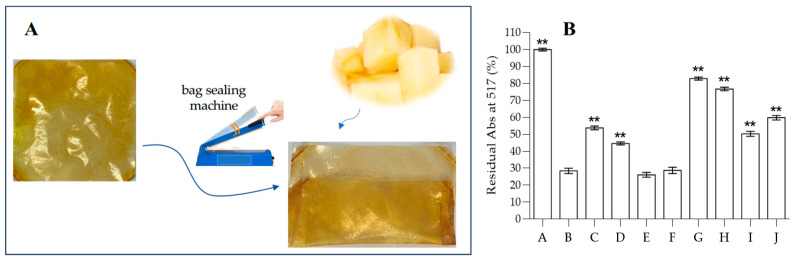
Schematic representation of the bioplastic bag preparation by heat-sealing method (**A**) and analysis of the changes in the antioxidant activity (**B**) of apple samples packed or not packed after different intervals of time (0, 24 and 48 h). As far as the preparation of the bags are concerned, the bioplastic has been cut into square pieces (6.0 cm × 6.0 cm) and two sides have been sealed with a bag-sealing machine. Once the cubes of apple have been inserted, the last side has been sealed with a bag-sealing machine. The letters in the graph indicate the following: A, B and G) apple samples not packed; C and H) apple sample packed with PHA film; D and I) apple samples packed with PVA plus 0.1% flavonoid fraction and liquid fraction obtained after digestion with cellulase and pectinase; E and L) PVA plus 0.1% flavonoid fraction, liquid fraction obtained after digestion with cellulase and pectinase and 0.5 g of the solid material remaining after enzyme treatment; F and M) PVA plus 0.1% flavonoid fraction, liquid fraction obtained after digestion with cellulase and pectinase and 1.0 g of the solid material remaining after enzyme treatment. The ** indicates significant changes with respect to apple samples not packed at 0 h at *p* > 0.05.

**Table 1 ijms-26-09470-t001:** Thickness, tensile parameters and contact angle of PVA alone or modified by the addition of the recycling elements obtained from *Citrus bergamia* by-product. (A) PVA alone; (B) PVA plus 0.1% flavonoid fraction and liquid fraction obtained after digestion with cellulase and pectinase; (C) PVA plus 0.1% flavonoid fraction, liquid fraction obtained after digestion with cellulase and pectinase and 0.5 g of the solid material remaining after enzyme treatment; (D) PVA plus 0.1% flavonoid fraction, liquid fraction obtained after digestion with cellulase and pectinase and 1.0 g of the solid material remaining after enzyme treatment.

Biofilms	Thickness [μm]	E [MPa]	σ_b_ [MPa]	ε_β_ [%]	θw [°]
A	32.01 ± 10.07	1221.69 ± 5.34	35.77 ± 1.40	118.89 ± 6.29	14.91 ± 1.80
B	79.86 ± 10.50	526.26 ± 9.66	79.56 ± 2.21	384.16 ± 12.18	16.87 ± 2.83
C	78.05 ± 9.29	623.74 ± 8.04	20.14 ± 2.24	45.77 ± 10.02	21.70 ± 2.11
D	67.16 ± 10.80	594.04 ± 7.33	25.76 ± 1.11	50.91 ± 10.49	18.46 ± 2.07

**Table 2 ijms-26-09470-t002:** Colorimetric analysis of the PVA films with and without the modification due to the addition of the element obtained from *Citrus bergamia* by-product recycling process utilizing the ScieLAB scale. (A) PVA alone; (B) PVA plus 0.1% flavonoid fraction and liquid fraction obtained after digestion with cellulase and pectinase; (C) PVA plus 0.1% flavonoid fraction, liquid fraction obtained after digestion with cellulase and pectinase and 0.5 g of the solid material remaining after enzyme treatment; (D) PVA plus 0.1% flavonoid fraction, liquid fraction obtained after digestion with cellulase and pectinase and 1.0 g of the solid material remaining after enzyme treatment.

Functionalized Bioplastics		L*	a*	b*	ΔE*
**A**	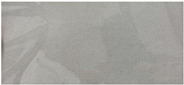	90.44 ± 0.16	−1.06 ± 0.02	2.77 ± 0.01	90.49 ± 0.16
**B**	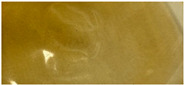	75.54 ± 1.24	3.85 ± 1.12	47.46 ± 3.24	89.35 ± 0.73
**C**	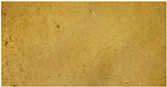	77.05 ± 1.59	2.70 ± 0.92	34.80 ± 2.80	84.63 ± 0.51
**D**	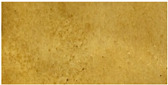	78.27 ± 2.57	2.31 ± 1.19	35.84 ± 3.04	86.93 ± 0.73

**Table 3 ijms-26-09470-t003:** Evaluation of the optical properties of the PVA films with and without the modification due to the addition of the element obtained from the *Citrus* waste recycling process. (A) PVA alone; (B) PVA plus 0.1% flavonoid fraction and liquid fraction obtained after digestion with cellulase and pectinase; (C) PVA plus 0.1% flavonoid fraction, liquid fraction obtained after digestion with cellulase, and pectinase and 0.5 g of the solid material remaining after enzyme treatment; (D) PVA plus 0.1% flavonoid fraction, liquid fraction obtained after digestion with cellulase and pectinase and 1.0 g of the solid material remaining after enzyme treatment.

Functionalized Bioplastics	T%	Opacity (Abs/mm^−1^)
A	89.75667 ± 0.499433	1.564
B	60.1075 ± 2.753983	3.158
C	32.974 ± 4.532878	6.883
D	38.292 ± 3.265803	6.222

**Table 4 ijms-26-09470-t004:** Evaluation of the color changes in the apple samples not packed or packed with PVA alone (A), PVA plus 0.1% flavonoid fraction and liquid fraction obtained after digestion with cellulase and pectinase (B), PVA plus 0.1% flavonoid fraction, liquid fraction obtained after digestion with cellulase and pectinase and 0.5 g of the solid material remaining after enzyme treatment (C), PVA plus 0.1% flavonoid fraction, liquid fraction obtained after digestion with cellulase and pectinase and 1.0 g of the solid material remaining after enzyme treatment (D) at different intervals of time (0, 24 and 48 h). Different superscript letters indicate a statistical significance difference at a *p* < 0.05 in the not packed samples after 0, 24 and 48 h. Different superscript numbers indicate a statistical significance difference at a *p* < 0.05 with respect to not packed samples after 24 or 48 h.

Hours		L*	a*	b*	ΔE*
Apple samples not packed	0	71.38 ± 1.31 ^a^	−1.61 ± 0.31 ^a^	19.24 ± 1.21 ^a^	73.94 ± 1.48 ^a^
Apple samples not packed		42.77 ± 1.08 ^b,1^	10.61 ± 0.02 ^b,1^	31.65 ± 3.96 ^b,1^	50.73 ± 1.01 ^b,1^
A	24	62.15 ± 4.23 ^2^	2.72 ± 0.88 ^2^	26.92 ± 2.91 ^2^	67.80 ± 4.92 ^2^
B		64.58 ± 2.82 ^2^	3.35 ± 2.12 ^2^	26.31 ± 2.45 ^2^	69.85 ± 3.11 ^2^
C		61.94 ± 4.79 ^2^	3.97 ± 1.00 ^2^	29.29 ± 0.88 ^2^	68.64 ± 4.76 ^2^
D		61.74 ± 4.25 ^2^	3.57 ± 1.20 ^2^	29.02 ± 0.79 ^2^	67.94 ± 4.37 ^2^
Apple samples not packed		42.60 ± 0.66 ^b,1^	15.12 ± 1.10 ^c,1^	34.82 ± 0.91 ^b,1^	57.07 ± 0.64 ^c,1^
A	48	46.89 ± 0.61 ^1^	14.67 ± 0.77 ^1^	38.99 ± 3.15 ^2^	62.75 ± 2.17 ^2^
B		61.76 ± 0.20 ^2^	11.64 ± 1.57 ^2^	39.04 ± 5.86 ^2^	74.11 ± 2.87 ^2^
C		55.62 ± 0.98 ^2^	13.79 ± 0.33 ^2^	42.63 ± 1.55 ^2^	71.42 ± 1.75 ^2^
D		55.13 ± 0.87 ^2^	13.57 ± 0.28 ^2^	42.22 ± 1.37 ^2^	71.32 ± 1.62 ^2^

## Data Availability

Not applicable.

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
