# Peer review of "Biochemical Modification of Poly-Vinyl-Alcohol-Based Bioplastics with Citrus By-Product to Increase Its Food Packaging Application"

_ijms, 2025, doi:10.3390/ijms26199470_

Round 1

Reviewer 1 Report

Comments and Suggestions for Authors

This article provides rich information about the modification of degradable bioplastic PVA for using in the field of food packaging. This is an interesting and valuable study. However, the following corrections are required before considering the article for publication.

  1. Page 10: The expression of “…if also the remaining solid fraction (obtained after the separation of the flavonoids and the enzymatic digestion) is added to the PVA, the mechanical resistance and the deformability of the material drastically worsen (to a value of about 20 MPa and of 45%, respectively)” and “This clearly indicates a decrease of the hydrophilicity of the starting material, making its utilization suitable with sample containing water.” seems like contradict with the express of “The functionalized produced materials show interesting results acquiring new properties (such as antioxidant activity) without drastically changing mechanical and hydrophilicity features” in conclusion. Please check them and rephrase these words.

  1. It is recommended to add a brief sample name next to the colorful lines in Figure 3 to distinguish them easily, when readers print this article in black and white. It is similar in Figure 5, and the illustration should be placed under the bottom of Figure 5.

  1. It is suggested to add the numbers and scales in the SEM images, which is convenient for readers to easily distinguish the differences between each picture.

  1. The information in Figure 8A is not clearly presented. It is suggested to add some relevant pictures, parameter descriptions or other relative information during the preparation process to make it clear at a glance.

  1. There are some formation and normative problems in the text. Please check the whole manuscript. Such as, some of the units in the text are written with symbols and some are written in English words, and it is recommended to keep uniform.

Page 9 Line 246: “32 microns to a value between 67 and 79 microns”, however, it is μm in Table 1. In addition, what does qw stand for in Table 1?

The written of g or gram, H, hours, or h is not uniform in the whole text.

The font of text in Figure 7 should be enlarged.

All formulas in the text should be numbered, especially,

3.6.4 Page 18 Line 562: The formula writing format is not standardized, and the meaning and unit of each symbol need to be explained in all formulas. Please check.

The written of a in Table 4 and α in the formula in Page 18 Line 569 is inconsistency.

The format of the references is not uniform, some references are incomplete, it is inconvenient to find the original text. It is recommended to carefully check and unify the content and format of each reference.

Author Response

This article provides rich information about the modification of degradable bioplastic PVA for using in the field of food packaging. This is an interesting and valuable study. However, the following corrections are required before considering the article for publication.

  1. Page 10: The expression of “…if also the remaining solid fraction (obtained after the separation of the flavonoids and the enzymatic digestion) is added to the PVA, the mechanical resistance and the deformability of the material drastically worsen (to a value of about 20 MPa and of 45%, respectively)” and “This clearly indicates a decrease of the hydrophilicity of the starting material, making its utilization suitable with sample containing water.” seems like contradict with the express of “The functionalized produced materials show interesting results acquiring new properties (such as antioxidant activity) without drastically changing mechanical and hydrophilicity features” in conclusion. Please check them and rephrase these words.

According to reviewer suggestion, we have rephrase this sentence

  1. It is recommended to add a brief sample name next to the colorful lines in Figure 3 to distinguish them easily, when readers print this article in black and white. It is similar in Figure 5, and the illustration should be placed under the bottom of Figure 5.

Done

  1. It is suggested to add the numbers and scales in the SEM images, which is convenient for readers to easily distinguish the differences between each picture.

According to reviewer suggestion we have made the change

  1. The information in Figure 8A is not clearly presented. It is suggested to add some relevant pictures, parameter descriptions or other relative information during the preparation process to make it clear at a glance.

According to reviewer suggestion we have added some picture and other information as far as the preparation are concerned. 

  1. There are some formation and normative problems in the text. Please check the whole manuscript. Such as, some of the units in the text are written with symbols and some are written in English words, and it is recommended to keep uniform.

Done.

Page 9 Line 246: “32 microns to a value between 67 and 79 microns”, however, it is μm in Table 1. In addition, what does qw stand for in Table 1?

Thank you for the suggestion, accordingly we uniformed in the test changing the word “microns” in “mm”, as requested ( see pag. 9/24). We corrected the symbols in table 1, and in particular “qw” that stands for “qw” the contact angle.

The written of g or gram, H, hours, or h is not uniform in the whole text.

Done

The font of text in Figure 7 should be enlarged.

Done

All formulas in the text should be numbered, especially,

3.6.4 Page 18 Line 562: The formula writing format is not standardized, and the meaning and unit of each symbol need to be explained in all formulas. Please check.

Thank you for the useful suggestion the element has been explained

The written of a in Table 4 and α in the formula in Page 18 Line 569 is inconsistency.

Thank you for the useful suggestion the element has been changed

The format of the references is not uniform, some references are incomplete, it is inconvenient to find the original text. It is recommended to carefully check and unify the content and format of each reference.

Done

Reviewer 2 Report

Comments and Suggestions for Authors

This manuscript focuses on modifying polyvinyl alcohol (PVA)-based bioplastics using three components from Citrus bergamia by-product - flavonoid fraction, enzymatic hydrolysate, and enzyme-treated solid residue with the aim of enhancing their applicability in food packaging. Aligned with the global demands for circular economy and biodegradable active packaging, the research direction is theoretically and practically relevant. The study adopts green extraction protocols, combines multiple characterization techniques (ATR-FTIR, SEM, XRD, etc.), and conducts functional validations (antioxidant activity, migration tests, apple preservation assays) to initially confirm the "active packaging" potential of the modified bioplastics. However, the manuscript exhibits shortcomings in experimental detail completeness, quantitative result analysis, discussion depth, and format standardization, which need to be addressed to improve the study’s scientific rigor and reproducibility.

1 In Section 3.2, the "pastazzo pretreatment" mentions "passed through a sieve with 100 μM mesh," which contains a unit error (μM refers to molar concentration; the correct unit should be μm). Additionally, the particle size distribution (e.g., D50, D90) of sieved samples and specific freeze-drying conditions (temperature, vacuum degree, time) are not provided - these parameters directly affect subsequent extraction efficiency and film uniformity. In Section 3.5, the "solvent casting method" for PVA film preparation lacks descriptions of drying conditions (temperature, relative humidity, time), which can lead to variations in film crystallinity (XRD results) and mechanical properties (e.g., tensile strength) when reproducing the experiment in other laboratories.

2 SEM analysis (Section 2.2.2) only describes "increased surface roughness and decreased porosity" but does not use tools to calculate quantitative indicators, making it impossible to establish a clear correlation between solid residue addition and surface morphology. XRD analysis (Section 2.2.3) notes "new peaks at 2θ=13° and 24.4°" but does not identify the corresponding crystal phases (e.g., cellulose type I/II, pectin characteristic peaks) or calculate the crystallinity index (via the Segal method), preventing a precise assessment of how modified components affect PVA’s crystalline structure

3  Section 2.5 claims that "adding solid residue reduces mechanical properties (tensile strength from 79 MPa to 20 MPa) due to plasticization," but no direct evidence is provided. The inference of "plasticization effect" based solely on tensile data is insufficiently persuasive. Section 2.6 only tests light transmittance at 600 nm for "UV barrier performance" and omits data for the UV-B region (280–320 nm) the key wavelength range causing food lipid oxidation and vitamin degradation—making it impossible to confirm the material’s practical light protection effect on food .

4 Reference formatting is inconsistent, requiring standardization. Some Reference (Polymer, 2025, 316, 127827; Polymer, 2025, 324, 128219) may be added in the the manuscript.

Author Response

This manuscript focuses on modifying polyvinyl alcohol (PVA)-based bioplastics using three components from Citrus bergamia by-product - flavonoid fraction, enzymatic hydrolysate, and enzyme-treated solid residue with the aim of enhancing their applicability in food packaging. Aligned with the global demands for circular economy and biodegradable active packaging, the research direction is theoretically and practically relevant. The study adopts green extraction protocols, combines multiple characterization techniques (ATR-FTIR, SEM, XRD, etc.), and conducts functional validations (antioxidant activity, migration tests, apple preservation assays) to initially confirm the "active packaging" potential of the modified bioplastics. However, the manuscript exhibits shortcomings in experimental detail completeness, quantitative result analysis, discussion depth, and format standardization, which need to be addressed to improve the study’s scientific rigor and reproducibility.

1 In Section 3.2, the "pastazzo pretreatment" mentions "passed through a sieve with 100 μM mesh," which contains a unit error (μM refers to molar concentration; the correct unit should be μm). Additionally, the particle size distribution (e.g., D50, D90) of sieved samples and specific freeze-drying conditions (temperature, vacuum degree, time) are not provided - these parameters directly affect subsequent extraction efficiency and film uniformity. In Section 3.5, the "solvent casting method" for PVA film preparation lacks descriptions of drying conditions (temperature, relative humidity, time), which can lead to variations in film crystallinity (XRD results) and mechanical properties (e.g., tensile strength) when reproducing the experiment in other laboratories.

Thank you for the very useful observation, and, accordingly, the requested information has been added.

2 SEM analysis (Section 2.2.2) only describes "increased surface roughness and decreased porosity" but does not use tools to calculate quantitative indicators, making it impossible to establish a clear correlation between solid residue addition and surface morphology. XRD analysis (Section 2.2.3) notes "new peaks at 2θ=13° and 24.4°" but does not identify the corresponding crystal phases (e.g., cellulose type I/II, pectin characteristic peaks) or calculate the crystallinity index (via the Segal method), preventing a precise assessment of how modified components affect PVA’s crystalline structure

We thank the reviewer for this comment. In the revised version, SEM images were quantitatively analyzed using ImageJ. Surface porosity values were quantified and expressed as percentages, whereas roughness parameters (Ra, Rq), provided by the software in arbitrary units, were employed exclusively to establish the relative trend among the samples. A comprehensive roughness characterization by AFM will be addressed in a forthcoming study. These data, together with the crystallinity index obtained from XRD (65% for neat PVA, 41–45% for samples 2–3, 60% for sample 4, with new peaks at 2θ ≈ 13° and 24° assigned to cellulose II), were added to the manuscript. The results quantitatively confirm the correlation between morphological/structural modifications and the mechanical and barrier performance of the films. In the revised manuscript we added the following text and image incorresponding the section:

“The SEM images, further processed into 3D surface plots (Figure 5), highlighted the effect of citrus-derived fractions on the topography of PVA-based films. The pure PVA film (sample 1) exhibited a relatively compact and homogeneous surface with moderate roughness and limited irregularities, consistent with a calculated surface porosity of 2.93%. Upon incorporation of the flavonoid-rich fraction and the liquid digested fraction (sample 2) and 0.5 mg solid fraction (samples 3), the overall topography appeared more irregular, with an increased number of surface protrusions and valleys. Interestingly, quantitative image analysis (with ImageJ) revealed that surface porosity decreased to 0.014% and 0.008% for samples 2 and 3, respectively, suggesting that these fractions compact or fill open pores despite increasing roughness. The lowest value of 0.005% was observed for the film containing the enzymatically digested solid residue (sample 4), despite its more pronounced surface roughness and heterogeneity. This apparent discrepancy can be explained by the fact that the incorporation of citrus-derived solids tends to partially fill or compact open surface pores, thereby reducing the two-dimensional porosity detected by threshold-based image analysis, while simultaneously generating localized agglomerates that emerge as surface peaks and enhance roughness. As a consequence, samples 2–4 show reduced planar porosity but increased morphological irregularities. These features correlate with the functional properties described in the preparation of the bioplastics: films with flavonoid and liquid digested fractions (samples 2) display improved tensile strength and barrier properties, whereas the excessive roughness and local discontinuities introduced by the maximum addition of solid residue (sample 4) act as structural modifier, leading to diminished mechanical resistance and higher opacity. The combined 3D surface and porosity analysis provide quantitative confirmation of the link between morphological modifications and the balance between strengthening and weakening effects in functionalized PVA films.

Figure 5. 3D surface plots of PVA-based films obtained from SEM micrographs after ImageJ pro-cessing: (1) PVA alone (sample 1); (2) PVA plus 0.1% flavonoid fraction and liquid fraction obtained after digestion with cellulase and pectinase (sample 2); (3) PVA plus 0.1% flavonoid fraction, liquid fraction obtained after digestion with cellulase and pectinase and 0.5 g of the solid material remain after enzyme treatment (sample 3); (4) PVA plus 0.1% flavonoid fraction, liquid fraction obtained after digestion with cellulase and pectinase and 1.0 g of the solid material remain after enzyme treatment (sample 4). A progressive decrease in surface porosity (2.93%, 0.014%, 0.008%, and 0.005% respectively) is observed, together with an increase in surface roughness and heterogeneity, particularly in sample 4, where the incorporation of solid residue leads to pronounced topographical irregularities.

Roughness parameters (Ra, Rq) were extracted from SEM line profiles using ImageJ and expressed in arbitrary intensity units, serving only as comparative indicators. The trend observed was Sample 4 < Sample 1 < Sample 2 < Sample 3. Interestingly, SEM line-profile analysis indicated lower Ra/Rq values for Sample 4, despite its irregular surface morphology. This apparent discrepancy arises from the method: single or limited line profiles capture average intensity variations along selected traces and may overlook sparse protrusions evident in 3D reconstructions. In fact, Sample 4, with very low porosity (0.005%), displays a morphology of few large agglomerates surrounded by smooth areas; as a result, line-based Ra appears lower, whereas area-based and 3D analyses reveal pronounced local irregularities.

(3.2.3) The morphological and structural modifications observed by SEM and XRD correlate directly with the functional properties of the films. Pure PVA exhibited the character-istic reflection at 2θ ≈ 20° with a crystallinity index (CI) of 65%. After incorporation of flavonoid and liquid digested fractions (sample 2) and the lowest amount of solid frac-tion (sample 3), the CI decreased to 41% and 45%, while simultaneously reducing sur-face porosity to 0.014% and 0.008%. This structural rearrangement promoted a more amorphous matrix and limited diffusion pathways, thereby enhancing chain mobility and enabling stronger intermolecular interactions between PVA and citrus-derived polysaccharides. As a result, these samples exhibited significantly improved tensile strength (up to 79.56 ± 2.21 MPa) and elongation (up to 384.16%), together with an in-creased contact angle (from ~15° in neat PVA to ~22°), reflecting improved barrier properties against aqueous environments. Conversely, the film containing the solid residue (sample 4) partially recovered crystallinity (CI = 60%), due to the presence of cellulose II X1, as demonstrated by the new peaks at 2θ ≈ 13° and 24° with index (110) and (020), respectively. The coexistence of crystalline cellulose domains and a hetero-geneous PVA network explains the increased surface irregularities observed in SEM, with surface porosity reduced to 0.005% but roughness markedly increased. These morphological discontinuities acted as stress concentrators, leading to decreased me-chanical resistance (20.14 ± 2.24 MPa; elongation 45.77%) and reduced transparency due to enhanced light scattering.

X1: Gong, Jie & Li, Jun & Xu, Jun & Xiang, Zhouyang & Mo, Lihuan. (2017). Research on cellulose nanocrystals produced from cellulose sources with various polymorphs. RSC Adv. 7. 33486-33493. 10.1039/C7RA06222B.”

3  Section 2.5 claims that "adding solid residue reduces mechanical properties (tensile strength from 79 MPa to 20 MPa) due to plasticization," but no direct evidence is provided. The inference of "plasticization effect" based solely on tensile data is insufficiently persuasive. Section 2.6 only tests light transmittance at 600 nm for "UV barrier performance" and omits data for the UV-B region (280–320 nm) the key wavelength range causing food lipid oxidation and vitamin degradation—making it impossible to confirm the material’s practical light protection effect on food.

As is known, the static tensile test evaluates the stiffness of a material through the analysis of the elastic modulus (E). In the case of our materials, it was not possible to add further investigations (measurement of the Shore D hardness) due to the excessive thinness of the film. Therefore, to better highlight the plasticizing effect, we added the stress-strain curves within the paper (see new figure 6).As far as the second point are concerned, we thank you for the useful suggestion, but we just utilized the reference test to analyse the opacity and, as reported in literature and correlate it with food protection. We think that your observation is correct and we delete the relative sentence in the main text.

4 Reference formatting is inconsistent, requiring standardization. Some Reference (Polymer, 2025, 316, 127827; Polymer, 2025, 324, 128219) may be added in the the manuscript.

Done

Reviewer 3 Report

Comments and Suggestions for Authors

The study deals with the PVA-alcohol bioplastics with Citrus by-product using biochemical modification to increase its food packaging application.

The study is timely and relevant. But several areas require clarification to enhance clarity, and reader engagement. I recommended publishing this work after the authors need to cover the following major revisions.

For English Writing and Style:

 - I do not feel qualified to judge the English language and style.

For the Introduction section:

  • The introduction is somewhat broad and could be more focused on the specific knowledge gap: What is missing in prior PVA-based bioplastic research?
  • Please cite recent works, if any, that report citrus waste valorization in active packaging.
  • Define explicitly the term (Class III bioplastics – in the conclusion section) as it may not be familiar to readers.

For the result section:

  • Extraction results: While AGREE program is an innovative tool, please report quantitative yields of polyphenols across extraction methods (reproducibility).
  • SEM/ XRD: The discussion of surface roughness and crystallinity is brief. Please explain how these modifications correlate with improved mechanical or barrier properties.
  • Antioxidant assays: Is there any available data of antioxidant capacity for commercial antioxidants or biopolymer systems to compare with to add to the product significancy.
  • Migration tests: You mentioned that ethanol caused the greatest release. Please provide discussion on implications.
  • Food self-life test: for 48 hours, is very limited. Longer term like 7 days would give stronger case for real packaging applications.
  • Table 4 missing statistical significance markers also Figure 7.

For the characterization section:

  • For FTIR results, please assign specific peaks to functional groups arising from flavonoid engagement.
  • The mechanical properties are reported but comparisons with conventional plastics may provide context.
  • I would recommend you link opacity and transparency results to food preservation benefits.

For the materials and methods section:

  • Some subsections are overly descriptive such as exact equipment suppliers. Consider simplifying to improve readability.
  • Clarify replicates and sample sizes is key. Make sure it is consistent across experiments.
  • Statistical methods are well-described but confidence intervals would improve interpretation.

For the conclusion section:

  • The conclusion is a repetitive of results, you may improve it by adding value to scaling challenges for industrial processing of citrus waste or regulatory pathways for commercialization.
  • You may add one or two sentences on future research directions such as self-life conditions.

Author Response

The study deals with the PVA-alcohol bioplastics with Citrus by-product using biochemical modification to increase its food packaging application.

The study is timely and relevant. But several areas require clarification to enhance clarity, and reader engagement. I recommended publishing this work after the authors need to cover the following major revisions.

For English Writing and Style:

 - I do not feel qualified to judge the English language and style.

For the Introduction section:

The introduction is somewhat broad and could be more focused on the specific knowledge gap: What is missing in prior PVA-based bioplastic research?

Thank you for the suggestion. We have tried to add some elements to highlight this point, but we do not find a lot of things and, for us it’s better to point out the potentiality to utilize all the by-product, transforming it from a cost to a resource.

Please cite recent works, if any, that report citrus waste valorization in active packaging.

As far as our knowledge are concerned, until now, no works are present in literature on “pastazzo” but we found few works that utilize the peel, and we added a sentence in the main text and relative citations:

 “Although recently, Citrus peel has been employed in the production of “packaging system” for protecting perishable fruits  because they are rich, at the same time, of cellulose, pectin and polyphenols, but still lack the utilization of the main by-product (“pastazzo”) derived from agro-industrial utilization []”

Shikai Zhang, Xinxin Cheng, Wenjing Yang, Quanbin Fu, Feng Su, Peng Wu, Yijing Li, Fen Wang, Houshen Li, Shiyun Ai, Converting fruit peels into biodegradable, recyclable and antimicrobial eco-friendly bioplastics for perishable fruit preservation, Bioresource Technology, Volume 406,2024,131074,ISSN 0960-8524. https://doi.org/10.1016/j.biortech.2024.131074.

Jayachandra S. Yaradoddi, Nagaraj R. Banapurmath, Sharanabasava V. Ganachari, Manzoore Elahi M. Soudagar, Ashok M. Sajjan, Shrinidhi Kamat, M.A. Mujtaba, Ashok S. Shettar, Ali E. Anqi, Mohammad Reza Safaei, Ashraf Elfasakhany, Md Irfanul Haque Siddiqui, Masood Ashraf Ali, Bio-based material from fruit waste of orange peel for industrial applications, Journal of Materials Research and Technology, Volume 17, 2022, Pages 3186-3197, ISSN 2238-7854, https://doi.org/10.1016/j.jmrt.2021.09.016.

Oluwasina, O.O., Awonyemi, I.O. Citrus Peel Extract Starch-Based Bioplastic: Effect of Extract Concentration on Packed Fish and Bioplastic Properties. J Polym Environ 29, 1706–1716 (2021). https://doi.org/10.1007/s10924-020-01990-7

Define explicitly the term (Class III bioplastics – in the conclusion section) as it may not be familiar to readers.

Done

For the result section:

Extraction results: While AGREE program is an innovative tool, please report quantitative yields of polyphenols across extraction methods (reproducibility).

Done.

SEM/ XRD: The discussion of surface roughness and crystallinity is brief. Please explain how these modifications correlate with improved mechanical or barrier properties.

In the revised manuscript we expanded the discussion of how morphological and structural modifications influence the mechanical and barrier performance of the films. SEM analysis showed a strong reduction in surface porosity (from 2.93% in neat PVA to 0.014–0.005% in functionalized films), while XRD revealed a decrease in crystallinity index for samples 2 and 3 (41–45%), consistent with a more amorphous structure. These combined effects favor stronger intermolecular interactions between PVA and citrus-derived polysaccharides, leading to increased tensile strength, elongation, and contact angle. Conversely, the partial recovery of crystallinity (60%) and the presence of cellulose II in sample 4 correlated with morphological heterogeneity, which acted as stress concentrators and reduced mechanical resistance. This extended discussion has been incorporated into the manuscript:

“The morphological and structural modifications observed by SEM and XRD correlate directly with the functional properties of the films. Pure PVA exhibited the character-istic reflection at 2θ ≈ 20° with a crystallinity index (CI) of 65%. After incorporation of flavonoid and liquid digested fractions (sample 2) and the lowest amount of solid frac-tion (sample 3), the CI decreased to 41% and 45%, while simultaneously reducing sur-face porosity to 0.014% and 0.008%. This structural rearrangement promoted a more amorphous matrix and limited diffusion pathways, thereby enhancing chain mobility and enabling stronger intermolecular interactions between PVA and citrus-derived polysaccharides. As a result, these samples exhibited significantly improved tensile strength (up to 79.56 ± 2.21 MPa) and elongation (up to 384.16%), together with an in-creased contact angle (from ~15° in neat PVA to ~22°), reflecting improved barrier properties against aqueous environments. Conversely, the film containing the solid residue (sample 4) partially recovered crystallinity (CI = 60%), due to the presence of cellulose II X1, as demonstrated by the new peaks at 2θ ≈ 13° and 24° with index (110) and (020), respectively. The coexistence of crystalline cellulose domains and a hetero-geneous PVA network explains the increased surface irregularities observed in SEM, with surface porosity reduced to 0.005% but roughness markedly increased. These morphological discontinuities acted as stress concentrators, leading to decreased me-chanical resistance (20.14 ± 2.24 MPa; elongation 45.77%) and reduced transparency due to enhanced light scattering.”

Antioxidant assays: Is there any available data of antioxidant capacity for commercial antioxidants or biopolymer systems to compare with to add to the product significancy.

They are present in literature but they are not directly and simple comparable with our materials, because we do not use a pure compounds but complex matrices. If you want you can check another work published by this group to find some indication: S.F. Mirpoor, G.T. Patane, I. Corrado, C.V.L. Giosafatto, G. Ginestra, A. Nostro, A. Foti, P.G. Gucciardi, G. Mandalari, D. Barreca, T. Gervasi, C. Pezzella, Functionalization of Polyhydroxyalkanoates (PHA)-Based Bioplastic with Phloretin for Active Food Packaging: Characterization of Its Mechanical, Antioxidant, and Antimicrobial Activities, Int J Mol Sci 24(14) (2023).

Migration tests: You mentioned that ethanol caused the greatest release. Please provide discussion on implications.

Done.

Food self-life test: for 48 hours, is very limited. Longer term like 7 days would give stronger case for real packaging applications.

Thank you for the useful information, and we will utilize it for a future study but actually we cannot apply to our study.

Table 4 missing statistical significance markers also Figure 7.

 Done.

For the characterization section:

For FTIR results, please assign specific peaks to functional groups arising from flavonoid engagement.

Done.

The mechanical properties are reported but comparisons with conventional plastics may provide context.

Since it is difficult to compare conventional plastics produced from fossil fuels (as they degrade over thousands of years), it is preferable to make a comparison with other conventional plastics of biological origin, i.e., from renewable sources because these can undergo microbial degradation, like the PVA studied in this work.

Thus we added these sentences in the text:

“Polybutylene succinate (PBS) is an example of a bioderived and biodegradable plastic of scientific interest, as it is used commercially for various applications []. Considering its tensile mechanical parameters (modulus: 328 MPa, breaking strength: 37 MPa, and elongation at break: 354% [X]), we can conclude that pure PVA (sample A) is stiffer and less deformable than PBS, since its elastic modulus is approximately 1200 MPa and its breaking strength is approximately 118% (Table 1). However, the mechanical behavior of PBS is more similar to sample B, which extends its deformability by approximately 384%. This indicates that the plasticizing effect of the flavonoids and the liquid fraction obtained after digestion with cellulose/pectinase makes PVA more similar to PBS. Therefore, it would be interesting to conduct a future study on the PBS biopolymer matrix modified with citrus by-products.”

I would recommend you link opacity and transparency results to food preservation benefits.

 Done

For the materials and methods section:

Some subsections are overly descriptive such as exact equipment suppliers. Consider simplifying to improve readability.

Done

Clarify replicates and sample sizes is key. Make sure it is consistent across experiments.

Done

Statistical methods are well-described but confidence intervals would improve interpretation.

Done

For the conclusion section:

The conclusion is a repetitive of results, you may improve it by adding value to scaling challenges for industrial processing of citrus waste or regulatory pathways for commercialization.

You may add one or two sentences on future research directions such as self-life conditions.

Done.

Reviewer 4 Report

Comments and Suggestions for Authors
  1. Different alternative materials for food packaging have been developed, for example several can also be mentioned and briefly compared such as ZnO NPs, polymers and composites 1021/acsfoodscitech.2c00043; 10.1016/j.mattod.2022.01.022; doi.org/10.1007/s11434-010-4326-6. I believe that in this way a better description of the novelty can be achieved.
  2. I would like to recommend to improve the resolution of the images. Some of them are of a low quality and should be improved.
  3. In Fig. 3 the number are assigned to the bands, however, it is unclear which number refers to which particular band.
  4. I would request to provide all the data as average and standard deviation to reveal any significant differences between the groups.
  5. Generally, standard stress-strain curves should be provided first to confirm the calculated and presented in the Table data. I would also request to discuss the mechanisms for the observed changes of the values of Young’s modulus, which plays an important role in mechanical performance of the food packaging materials.
  6. It is known that PVA-based materials are biodegradable, thus, I would request to provide some more details on the service life time materials made of them.

Author Response

  1. Different alternative materials for food packaging have been developed, for example several can also be mentioned and briefly compared such as ZnO NPs, polymers and composites 1021/acsfoodscitech.2c00043; 10.1016/j.mattod.2022.01.022; doi.org/10.1007/s11434-010-4326-6. I believe that in this way a better description of the novelty can be achieved.

Thank for your suggestion. We have done this comparison by briefly mentioned some bioplastic obtained by other Citrus byproducts.

  1. I would like to recommend to improve the resolution of the images. Some of them are of a low quality and should be improved.

Done.

  1. In Fig. 3 the number are assigned to the bands, however, it is unclear which number refers to which particular band.

According to reviewer suggestion we added these information in the capture of the figure.

  1. I would request to provide all the data as average and standard deviation to reveal any significant differences between the groups.

Done

  1. Generally, standard stress-strain curves should be provided first to confirm the calculated and presented in the Table data. I would also request to discuss the mechanisms for the observed changes of the values of Young’s modulus, which plays an important role in mechanical performance of the food packaging materials.

Acccording to reviewer suggestion, we added these sentences in the text:

“The addition of the flavanol fraction and the liquid fraction obtained after digestion with cellulase and pectinase decreases the stiffness of the material (from 1221 MPa to 526 MPa) because it lubricates the macromolecular chains, facilitating their sliding and therefore their deformability. However, the addition of 0.5 grams and 1.0 grams of the solid material remaining after the enzymatic treatment increases the stiffness again to 623 MPa and 594 MPa due to the stiffening effect caused by the presence of the solid fraction, which hinders the sliding of the macromolecular chains within the chemical structure of the bioplastic.”

  1. It is known that PVA-based materials are biodegradable, thus, I would request to provide some more details on the service life time materials made of them.

The degradation of Polyvinyl Alcohol (PVA) changes significantly, ranging from a few days to months or even longer, depending on factors like its molecular weight, degree of hydrolysis, the specific environment (moist soil, dry soil, adapted wastewater sludge), and exposition to microbial population able to utilize as a carbon source. PVA, just for instance,  is biodagradate in about 3 days in moist soil, while in unadapted environments or in dry soil, degradation can take much longer (up to 6-9 months). 

Round 2

Reviewer 2 Report

Comments and Suggestions for Authors

Accept in present form

Comments on the Quality of English Language

Accept in present form

Reviewer 3 Report

Comments and Suggestions for Authors

Accept in its current form, as the authors have covered all my comments

Reviewer 4 Report

Comments and Suggestions for Authors

The manuscript was improved and could now be accepted without any additional corrections.